# What Planning Problems Can
# A Relational Neural Network Solve?

**Jiayuan Mao**[1]   **Tomás Lozano-Pérez**[1]   **Joshua B. Tenenbaum**[1,2,3]   **Leslie Pack Kaelbling**[1]

[1] MIT Computer Science & Artificial Intelligence Laboratory
[2] MIT Department of Brain and Cognitive Sciences
[3] Center for Brains, Minds and Machines

## Abstract

Goal-conditioned policies are generally understood to be "feed-forward" circuits, in the form of neural networks that map from the current state and the goal specification to the next action to take. However, under what circumstances such a policy can be learned and how efficient the policy will be are not well understood. In this paper, we present a circuit complexity analysis for relational neural networks (such as graph neural networks and transformers) representing policies for planning problems, by drawing connections with serialized goal regression search (S-GRS). We show that there are three general classes of planning problems, in terms of the growth of circuit width and depth as a function of the number of objects and planning horizon, providing constructive proofs. We also illustrate the utility of this analysis for designing neural networks for policy learning.

## 1   Introduction

Goal-conditioned policies are generally understood to be "feed-forward" circuits, in the form of neural networks such as multi-layer perceptrons (MLPs) or transformers [Vaswani et al., 2017]. They take a representation of the current world state and the goal as input, and generate an action as output. Previous work has proposed methods for learning such policies for particular problems via imitation or reinforcement learning [Wang et al., 2018, Dong et al., 2019, Li et al., 2020]; recently, others have tried probing current large-language models to understand the extent to which they already embody policies for a wide variety of problems [Liang et al., 2022, Carta et al., 2023].

However, a major theoretical challenge remains. In general, we understand that planning problems are PSPACE-hard with respect to the size of the state space [Bylander, 1994], but there seem to exist efficient (possibly suboptimal) policies for many problems such as block stacking [Dong et al., 2019] that generalize to arbitrarily sized problem instances. In this paper, we seek to understand and clarify the circuit complexity of goal-conditioned policies for classes of "classical" discrete planning problems: under what circumstances can a polynomial-sized policy circuit be constructed, and what is its size? Specifically, we highlight a relationship between a problem hardness measure (*regression width*, related to the notion of "width" in the forward planning literature [Chen and Giménez, 2007]) and circuit complexity. We concretely provide upper bounds on the policy circuit complexity as a function of the problem's regression width, using constructive proofs that yield algorithms for directly compiling a planning problem description into a goal-conditioned policy. We further show that such policies can be learned with conventional policy gradient methods from environment interactions only, with the theoretical results predicting the necessary size of the network.

There are several useful implications of these results. First, by analyzing concrete planning domains, we show that for many domains, there do exist simple policies that will generalize to problem instances with an arbitrary number of objects. Second, our theory predicts the circuit complexity of policies for these problems. Finally, our analysis suggests insights into why certain planning

---

Project page: https://concepts-ai.com/p/goal-regression-width/

37th Conference on Neural Information Processing Systems (NeurIPS 2023).

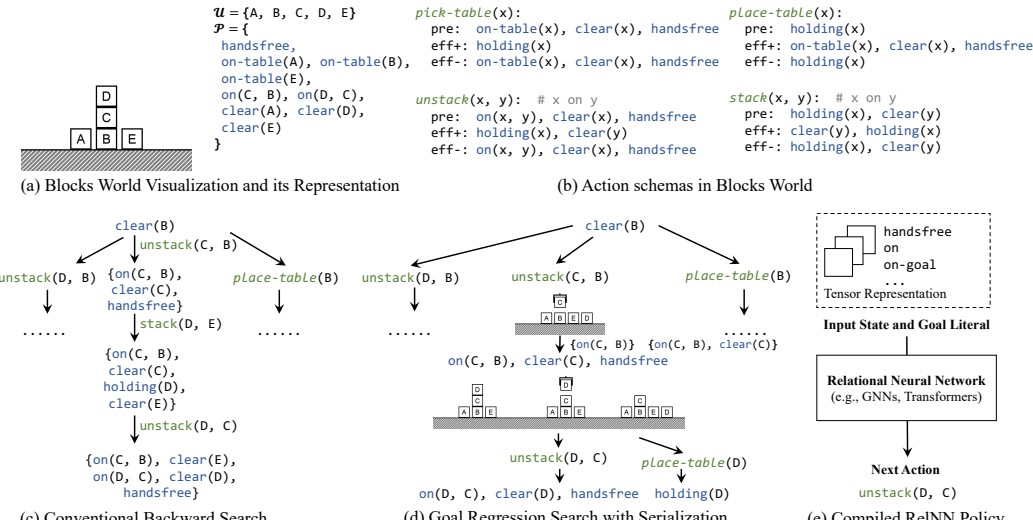

Figure 1: (a) Illustration of the Blocks World domain that we will be using as the main example. (b) The action schema definition in Blocks World. *clear*(*x*) means there is no object on *x*. (c) A backward search tree for solving the goal *clear*(B). (d) A serialized goal regression search tree for the same goal. (e) The form of a goal-conditioned policy for this problem.

problems, such as Sokoban and general task and motion planning (TAMP) problems, are hard, and likely cannot be solved *in general* by fixed-sized MLP or transformer-based policies [Culberson, 1997, Vega-Brown and Roy, 2020]. In the rest of the paper, Section 2 provides problem definitions, Sections 3 and 4 provides complexity definitions, theoretical results, and the policy compilation algorithm; Section 5 discusses the practical implications of these results.

## 2    Preliminaries: Planning Domain and Problem

Throughout this paper, we focus on analyzing the search complexity and policy circuit complexity of classical planning problems. Importantly, these planning problems have an object-centric representation and sparsity in the transition models: the state of the world is represented as a set of entities and their properties and relationships, while each action only changes a sparse set of properties and relations of a few objects. These features will contribute to search efficiency.

Formally, we consider the problem of planning and policy learning for a space $\mathcal{S}$ of world states. A planning problem is a tuple $\langle \mathcal{S}, s_0, \mathcal{G}, \mathcal{A}, \mathcal{T} \rangle$, where $s_0 \in \mathcal{S}$ is the initial state, $\mathcal{G} \subseteq \mathcal{S}$ is a goal specification, $\mathcal{A}$ is a set of actions that the agent can execute, and $\mathcal{T}$ is a (possibly partial) environmental transition model $\mathcal{T} : \mathcal{S} \times \mathcal{A} \to \mathcal{S}$. The task of planning is to output a sequence of actions $\bar{a}$ in which the terminal state $s_T$ induced by applying $a_i$ sequentially following $\mathcal{T}$ satisfies $s_T \in \mathcal{G}$. The task of policy learning is to find a function $\pi(s, \mathcal{G})$ that maps from the state and a goal specification to the next action so that applying $\pi$ recurrently on $s_0$ eventually yields a state $s_T \in \mathcal{G}$.

We use atomic STRIPS [Fikes and Nilsson, 1971, Lifschitz, 1987] to formalize the object-centric factorization of our planning problems. Specifically, as illustrated in Fig. 1a, the environmental state can be represented as a tuple $(\mathcal{U}_s, \mathcal{P}_s)$, where $\mathcal{U}_s$ is the set of objects in state $s$, denoted by arbitrary unique names (e.g., A, B). The second component, $\mathcal{P}_s$, is a set of *atoms*: *handfree*(), *on-table*(A), *on*(C, B), etc. Each atom contains a predicate name (e.g., *on*) and a list of arguments (e.g., C, B). All atoms in $\mathcal{P}_s$ are true at state $s$, while any atoms not in $\mathcal{P}_s$ are false. Since we do not consider object creation and deletion, we simply use $\mathcal{P}_0$ to denote the set of all possible atoms for the object universe $\mathcal{U}_{s_0}$*.

Illustrated in Fig. 1b, a (possibly partial) transition model $\mathcal{T}$ is specified in terms of a set of object-parameterized action schemas $\langle name, args, precond, effect \rangle$, where *name* is the name of the action, *args* is a list of symbols, *precond* and *effect* are descriptions of the action's effects. In the basic STRIPS formulation, both *precond* and *effect* are sets of atoms with variables in *args*, and *effect* is further decomposed into $eff_+$ and $eff_-$, denoting "add" effects and "delete" effects. An action schema (e.g., *unstack*) can be grounded into a concrete action $a$ (e.g., *unstack*(D, C)) by binding all variables in *args*.

---

*For simplicity, throughout the paper, we will be focusing on Boolean state variables.

| **Algorithm 1** Plain backward search. | **Algorithm 2** Serialized goal regression search. |
|---|---|
| **function** BWD($s_0$, $\mathcal{A}$, $goal\_set$) | **function** S-GRS($s_0$, $\mathcal{R}$, $g$, $cons$) |
|   **if** $goal\_set \subseteq s_0$ **then return** empty_list() |   **if** $g \in s_0$ **then return** ($s_0$) |
|   $possible\_t$ = empty_set() |   $possible\_t$ = empty_set() |
|   **for** $a \in \{a \in \mathcal{A} \mid goal\_set \cap eff\_(a) =$ |   **for** $r \in \{r \in \mathcal{R} \mid cons \cap eff\_(action(r)) = \emptyset$ **and** |
| $\emptyset$ **and** $goal\_set \cap eff_+ \neq \emptyset\}$ **do** | $goal(r) = g\}$ **do** |
|     $new\_goals = goal\_set \cup pre(a) \setminus eff_+(a)$ |     $p_1, p_2, \cdots, p_k$ = subgoals($r$) |
|     **if** $new\_goals \in goal\ stack$ **then continue** |     **if** $\exists p_i$ s.t. $p_i \in goal\ stack$ **then continue** |
|   |     **for** $i$ in $1, 2, \cdots, k$ **do** |
|   |       $new\_c = cons \cup \{p_1, \cdots, p_{i-1}\}$ |
|     $\pi$ = BWD($s_0$, $new\_goals$) |       $\pi_i$ = S-GRS($s_{i-1}$, $p_i$, $new\_c$) |
|   |       **if** $\pi_i == \perp$ **then break** |
|   |       $s_i = \mathcal{T}(s_{i-1}, \pi_i[-1])$ |
|     **if** $\pi \neq \perp$ **then** |     **if** $\pi_k \neq \perp$ **then** |
|       $possible\_t.add(\pi + \{a\})$ |       $possible\_t.add(\pi_1 + \ldots + \pi_k + \{a\})$ |
|   **return** shortest path in $possible\_t$ |   **return** shortest path in $possible\_t$ |

The formal semantics of STRIPS actions is: $\forall s. \forall a. pre(a) \subseteq s \implies (\mathcal{T}(s, a) = s \cup eff_+(a) \setminus eff\_(a))$. That is, for any state $s$ and any action $a$, if the preconditions of $a$ are satisfied in $s$, the state resulting from applying $a$ to $s$ will be $s \cup eff_+(a) \setminus eff\_(a)$. Note that $\mathcal{T}$ may not be defined for all $(s, a)$ pairs. Furthermore, we will only consider cases where the goal specification is a single atom, although more complicated conjunctive goals or goals that involve existential quantifiers can be easily supported by introducing an additional "goal-achieve" action that lists all atoms in the original goal as its precondition. From now on, we will use $g$ to denote the single goal atom of the problem.

# 3 Goal Regression Search and Width

A simple and effective way to solve STRIPS problems is backward search[†]. Illustrated in Algorithm 1[‡], we start from the goal $\{g\}$, and search for the last action $a$ that can be applied to achieve the goal (i.e., $g \in eff_+(a)$). Then, we add all the preconditions of action $a$ to our search target and recurse. To avoid infinite loops, we also keep track of a "*goal stack*" variable composed of all *goal_set*'s that have appeared in the depth-first search stack. The run-time of this algorithm has critical dependencies on (1) the number of steps in the resulting plan and (2) the number of atoms in the intermediate goal sets in search. In general, it has a worst-case time complexity exponential in the number of atoms.

## 3.1 Serialized Goal Regression Search

One possible way to speed up backward search is to *serialize* the subgoals in the goal set: that is, to commit to achieving them in a particular order. In the following sections, we study a variant of the plain backward search algorithm, namely *serialized goal regression search* (S-GRS, Algorithm 2), which uses a sequential decomposition of the preconditions: given the goal atom $g$, it searches for the last action $a$ and also an order in which the preconditions of $a$ should be accomplished. Of course, this method cannot change the worst-case complexity of the problem (actually, it may slow down the algorithm in cases where we need to search through many orders of the preconditions), but there are useful subclasses of planning problems for which it can achieve a substantial speed-up.

A *serialized goal regression rule* in $\mathcal{R}$, informally, takes the following form: $g \leftsquigarrow p_1, p_2, \cdots, p_k \parallel a$. It reads: in order to achieve $g$, which is a single atom, we need to achieve atoms $p_1, \cdots, p_k$ sequentially, and then execute action $a$. To formally define the notion of achieving atoms $p_1, \cdots, p_k$ "*sequentially*," we must include constraints in goal regression rules. Formally, we consider the application of a rule under constraint $c$ to have the following definition, $g^c \leftsquigarrow p_1^c, p_2^{c \cup \{p_1\}}, \cdots, p_k^{c \cup \{p_1, \cdots p_{k-1}\}} \parallel a$ which reads: in order to achieve $g$ without deleting atoms in set $c$, we can first achieve $p_1$ while maintaining $c$, then achieve $p_2$ while maintaining $c$ and $p_1$, $\ldots$, until we have achieved $p_k$. Finally, we perform $a$ to achieve $g$. For example, in Blocks World, we have the rule: $clear(B)^\emptyset \leftsquigarrow on(A, B), clear(A)^{\{on(A,B)\}}, handsfree()^{\{on(A,B),clear(A)\}} \parallel unstack(A, B)$. Note that many rules might be infeasible for a given state, because it is simply not possible to achieve $p_i$ while

---

[†]Fundamentally, backward search has the same worst-case time complexity as forward search. We choose to analyze backward search because goal regression rules are helpful in determining fine-grained search complexity.

[‡]We show the recursive version for clarity. This algorithm can also be implemented as a breadth-first search.

maintaining $c$ and $\{p_1, \cdots, p_{i-1}\}$. In this case, achieving *handsfree*() before *clear*(A) is infeasible because *handsfree*() may be falsified while achieving *clear*(A), if we need to move other blocks that are currently on A. Similarly, *on*(A, B) must be achieved before *clear*(A) because moving A onto B will break *clear*(A) (because we need to pick up A).

The set of all possible goal regression rules can be instantiated based on all ground actions $\mathcal{A}$ in a planning problem. Specifically, for a given atom $g$ (e.g., *holding*(A)) and a set of constraints $c$, and for each action $a \in \mathcal{A}$ (e.g., *pick-table*(A)), if $g \in \mathit{eff}_+(a)$ while $c \cap \mathit{eff}_-(a) = \emptyset$, then for any permutation of *pre*($a$), there will be a goal regression rule: $g^c \leftsquigarrow p_1, \cdots, p_k \,\|\, a$, where $p_1, \cdots, p_k$ is a permutation of *pre*($a$). We call this regression rule set $\mathcal{R}_0$.

Given a set of regression rules (e.g., $\mathcal{R}_0$), we can apply the serialized goal-regression search (S-GRS) algorithm, shown in Algorithm 2. S-GRS returns a shortest path from *state* to achieve *goal* while maintaining *cons*. Unfortunately, this algorithm is not optimal nor even complete (e.g., a sequence of preconditions may not be "serializable," which we will define formally later), in the general case. To make it complete, it is necessary to backtrack through different action sequences to achieve each subgoal, which increases time complexity. We include a discussion in Appendix A.1.

## 3.2 Serialization of Goal Regression Rules

Although Algorithm 2 is not complete in general, it provides insights about goal regression. In the following, we will introduce two properties of planning problems such that S-GRS becomes optimal, complete, and efficient. For brevity, we define *OptSearch*(*state*, *goal*, *cons*) as the set of optimal trajectories that achieve *goal*, from *state*, while maintaining *cons*. Here, *goal* can be a conjunction.

**Definition 3.1** (Optimal serializability). *A goal regression rule $g^{cons} \leftsquigarrow p_1, \cdots, p_k \,\|\, a$ is optimally serializable w.r.t. a state $s$ if and only if, for all steps $i$, if $\forall \pi_{<i} \in OptSearch(s, p_{<i} \wedge \cdots \wedge p_{i-1}, cons)$ and $\forall \pi_i \in OptSearch(\mathcal{T}(s, \pi_1), p_i, cons \cup \{p_1, \cdots, p_{i-1}\})$ then $\pi_{<i} \oplus \pi_i \in OptSearch(s, p_1 \wedge \cdots \wedge p_i, cons)$. Furthermore, $\forall \pi \in OptSearch(s, p_1 \wedge \cdots \wedge p_k, cons)$, $\pi \oplus a \in OptSearch(s, g, cons)$*[§].

Intuitively, a rule is optimally serializable if any optimal plan for the length $i - 1$ prefix of its preconditions can be extended into an optimal plan for achieving the length $i$ prefix. For example, in Blocks World, the rule *holding*(A) $\leftsquigarrow$ *on*(A, B), *clear*(A), *handsfree*() $\|$ *unstack*(A, B) is optimally serializable when *on*(A, B) is true for $s$. We define the set of optimally serializable rules for a state $s$ as *OSR*($s$), and the set of single-literal goals that can be solved with *OSR*(·) from $s_0$ as *OSG*($s_0$)[¶].

**Theorem 3.1.** *For any goal $g \in OSG(s_0)$, S-GRS is optimal and complete. See proof in Appendix A.2.*

## 3.3 Width of Search Problems

Chen and Giménez [2007] introduced the notion of the *width* of a planning problem and showed that the forward-search complexity is exponential in the width of a problem. We extend this notion to regression-based searches and introduce a generalized version of regression rules in which not all previously achieved preconditions must be explicitly maintained.

**Definition 3.2** (Generalized regression rules). *A generalized, optimally-serializable regression rule is $g^c \leftsquigarrow p_1^{c \cup c_1}, p_2^{c \cup c_2}, \cdots, p_k^{c \cup c_k} \,\|\, a$, where $c_i \subseteq \{p_1, p_2, \cdots, p_{i-1}\}$. Its width is $|c| + \max\{|c_1|, \cdots, |c_k|\}$.*

For example, for any initial state $s_0$ that satisfies *on*(A, B), the following generalized rule is optimally serializable: *clear*(B)$^\emptyset$ $\leftsquigarrow$ *on*(A, B)$^\emptyset$, *clear*(A)$^\emptyset$, *handsfree*()$^{\{clear(A)\}}$ $\|$ *unstack*(A, B). When we plan for the second precondition *clear*(A), we can ignore the first condition *on*(A, B), because the optimal plan for *clear*(A) will not change *on*(A, B). Using such generalized rules to replace the default rule set $\mathcal{R}_0$ improves search efficiency by reducing the number of possible subgoals. To define existence conditions for highly efficient search algorithms and policies, we need a stronger notion:

**Definition 3.3** (Strong optimally-serializable (SOS) width of regression rules). *A generalized regression rule has strong optimally-serializable width $k$ w.r.t. state $s$ if (a) it is optimally serializable, (b) its width is $k$, and, (c) $\forall c \, OptSearch(s, p_i \cup c_i \cup c, \emptyset) \subseteq OptSearch(s, \{p_1, \cdots, p_i\} \cup c, \emptyset)$.*

In the Blocks World domain, given a particular state $s$, if *on*(A, B) $\in s$, the generalized goal regression rule *clear*(B) $\leftsquigarrow$ *on*(A, B), *clear*(A), *handsfree*()$^{\{clear(A)\}}$ $\|$ *unstack*(A, B) is strong optimally-serializable, because the optimal trajectory to achieve *clear*(A) will not move A.

---

[§]$a \oplus b$ denotes concatenation of sequences. $\pi_{<i}$ denotes the subsequence $\{a_0, \cdots, a_{i-1}\}$

[¶]It is possible to make a non-optimally-serializable rule into an optimally serializable one by introducing "super predicates" as the conjunction of existing predicates, at the cost of increasing the state and action space sizes. We include discussions in Appendix A.3.

**Definition 3.4** (SOS width of problems)**.** *A planning problem $P = \langle \mathcal{S}, s_0, g, \mathcal{A}, \mathcal{T} \rangle$ has strong optimally-serializable width $k$ if there exists a set of strong optimally-serializable width $k$ rules $\mathcal{R}$ w.r.t. $s_0$, such that S-GRS can solve P using only rules in $\mathcal{R}$.*

**Theorem 3.2.** *If a problem P is of SOS width $k$, it can be solved in time $O(N^{k+1})$ with S-GRS, where N is the number of atoms. Here we have omitted polynomials related to enumerating all possible actions and their serializations (which are polynomial w.r.t. the number of objects in $\mathcal{U}$).*

*Proof idea.* We start by assuming knowing $\mathcal{R}$: the proof can be simply done by analyzing function calls: there are at most $O(N^{k+1})$ possible argument combinations of (*goal*, *cons*). Next, since the enumeration of all possible width-$k$ rules can be done in polynomial time. Therefore, the algorithm runs in polynomial time and it does not need to know $\mathcal{R}$ a priori. Note that the SOS condition cannot be removed because even if regression rules have a small width, there will be a possibly exponential branching factor caused by "free variables" in regression rules that do not appear in the goal atom. See the full proof in Appendix A.4. We also describe the connections with classical planning width and other related concepts, as well as how to generalize to $\forall$-quantified preconditions in Appendix A.5.

Importantly, planning problems such as Blocks World and Logistics have constant SOS widths, independent of the number of objects. Therefore, there exist polynomial-time algorithms for finding their solutions. Our analysis is inspired by and closely related to classical (forward) planning width Chen and Giménez [2007] and Lipovetzky and Geffner [2012]. Indeed, we have:

**Theorem 3.3.** *Any planning problem that has SOS-width $k$ has a forward width of at most $k + 1$ and, hence, can be solved by the algorithm IW($k + 1$). See the full proof in Appendix A.6.*

**Remark.** Unfortunately, SOS width and forward width are not exactly equivalent to each other. There exist problems whose forward width is smaller than $k + 1$, where $k$ is the SOS width. Appendix A.6 presents a concrete example. However, there are two advantages of our SOS width analysis over the forward width analysis. In particular, proofs for (forward) widths of planning problems were mostly done by analyzing solution structures. By contrast, our constructions form a new perspective: *a problem has a large width if the number of constraints to track during a goal regression search is large*. This view is helpful because our analyses can now focus on concrete regression rules for each individual operator in the domain — how its preconditions can be serialized. Second and more importantly, our "backward" view allows us to reason about circuit complexities of policies.

## 4 Policy Realization

We have studied sequential backward search algorithms for planning; now, we address the questions of how best to "compile" or "learn" them as feed-forward circuits representing goal-conditioned policies, and how the size of these circuits depends on properties of the problem. Here, we will focus on understanding the complexity of a certain *problem class*; that is, a set of problems in the same domain (e.g., Blocks World) with a similar goal predicate (e.g., having one particular block clear). In particular, we will make use of *relational neural networks* (*RelNN*, such as graph neural networks [Morris et al., 2019] and Transformers [Vaswani et al., 2017]). They accept inputs of arbitrary size, which is critical for building solutions to problems with arbitrary-sized universes. Another critical fact is that these networks have "parameter-tying" in the sense that there is a constant-sized set of parameters that is re-used according to a given pattern to realize a network.

In particular, as illustrated in Fig. 2, we will show how to take the set of operator descriptions for a planning domain and construct a relational neural network that represents a goal-conditioned policy: it takes a state and a goal (encoded as a special predicate, e.g., *on-goal*) and outputs a ground action. Here, the state is represented as a graph (objects are nodes, and relations are edges; possibly there will be hyperedges for high-arity relations). For example, the input contains nullary features (a single vector for the entire environment), unary features (a vector for each object), binary features (a vector for each pair of objects), etc. The goal predicate can be represented similarly. These inputs are usually represented as tensors. The complexity of this neural network will depend on the SOS width of problems in the problem class of interest.

We will develop two compilation strategies. First, in Section 4.2, we directly compile the BWD and the S-GRS algorithms into *RelNN* circuits. For problems with constant SOS width that is independent of the size of the universe, there will exist finite-breadth circuits (but the depth may depend on the size of the universe). In Section 4.3, we discuss a new property of planning problems—the existence of *regression rule selector* circuits—which results in policy circuits that are smaller than those generated

by previous strategies, and they are potentially of finite depth. The construction of the regression rule selector is not automatic and will generally require human priors or machine learning.

## 4.1 Relational Neural Network Background

We first quantify what a relational network can compute, using the formalization developed by Luo et al. [2022]; see also Morris et al. [2019], Barceló et al. [2020]. Let depth $D$ be the number of layers in a RelNN, and breadth $B$ be the maximum arity of the relations (hyperedges) in the network. For example, to represent a vector embedding for each tuple of size 5, $f(x_1, x_2, \cdots, x_5)$, where $x_1, \cdots, x_5 \in \mathcal{U}$ are entities in the planning problem, we need a relational neural network with breadth 5. We will only consider networks with a constant breadth. We denote the family of relational neural networks with depth $D$ and breadth $B$ as $RelNN[D, B]$. We will not be modeling the actual "hidden dimension" of the neural network layers (i.e., the number of neurons inside each layer), but we will assume that it is bounded (for

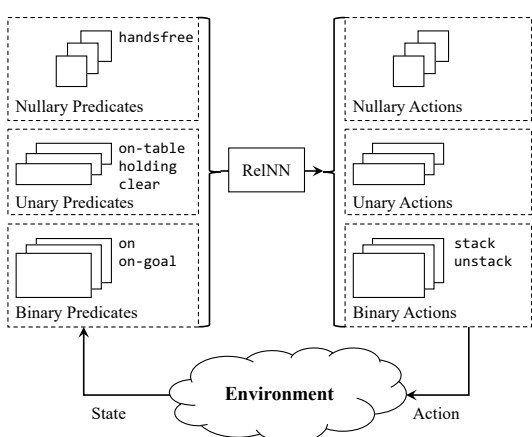

Figure 2: The input and output of a relational neural network (*RelNN*) policy.

example, as a function of $B$ and the number of predicates in the domain). Under such assumptions (most practical graph neural networks, Transformers, and hypergraph neural networks do follow these assumptions), we have the following lemma.

**Lemma 4.1** (Logical expressiveness of relational neural networks [Luo et al., 2022, Cai et al., 1992]). *Let $FOC_B$ denote a fragment of first-order logic with at most $B$ variables, extended with counting quantifiers of the form $\exists^{\geq_n} \phi$, which state that there are at least n nodes satisfying formula $\phi$.*

- *(Upper Bound) Any function $f$ in $FOC_B$ can be realized by $RelNN[D, B]$ for some D.*
- *(Lower Bound) There exists a function $f$ in $FOC_B$ such that for all D, $f$ cannot be realized by $RelNN[D, B-1]$.*

Here, we sketch a constructive proof for using RelNNs to realize FOL formulas. The breadth $B$ is analogous to the number of variables in FOL for encoding the value of the expression; the depth $D$ is the number of "nested quantifiers." For example, the formula $\exists x. \forall y. \forall z. p(x, y, z)$ needs 3 layers, one for each quantifier. Furthermore, we call a *RelNN* finite breadth (depth) if $B$ ($D$) is independent of the number of objects. Otherwise, we call it unbounded breadth (depth).

## 4.2 Compilation of BWD and S-GRS

Lemma 4.1 states an "equivalence" between the expressive power of relational neural networks and first-order logic formulas. In the following, we take advantage of this equivalence to compile search algorithms into relational neural networks. First, we have the following theorem.

**Theorem 4.1** (Compilation of BWD). *Given a planning problem P, let T be the length of the optimal trajectory (the planning horizon), $k_{BWD}$ be the maximum number of atoms in the goal set in BWD, and $\beta$ be the maximum arity of atoms in the domain. The backward search algorithm BWD can be compiled into a relational neural network in $RelNN[O(T), \beta \cdot k_{BWD}]$.*

*Proof sketch.* We provide a construction in which the *RelNN* computes a set of subgoals (i.e., a set of sets of ground atoms) $Goal^d$ at each layer $d$. Initially, $Goal^0 = \{\{g\}\}$. Then, $sg \in Goal^d$ if there is a path of length $d$ from any state $s$ that satisfies $sg \subseteq s$ to $g$. See Appendix B.1 for the full proof.

Although this construction is general and powerful, it is unrealistic for large problems because the depth can be exponential in the number of objects, and the number of atoms in the subgoal conjunctive formulas can be exponential in the depth. Therefore, in the following, we will leverage the idea of serialized goal regression to make a more efficient construction.

**Theorem 4.2** (Compilation of S-GRS). *Given a planning problem P of SOS width k, let T be the length of the optimal trajectory, and $\beta$ be the maximum arity of atoms in the domain. The serialized goal regression search S-GRS can be compiled into a $RelNN[O(T), (k+1) \cdot \beta]$, where $k + 1 \leq k_{BWD}$. See Appendix B.2 for the full proof using a similar technique as in Theorem 4.1.*

```
on-table(x) ⤳  if True: apply: holding(x) || place-table(x)
on(x, y)    ⤳  if True: apply: clear(y), holding(x) || stack(x, y)
holding(x)  ⤳  if exists y. on(x, y):
                   apply: on(x, y), clear(x), handsfree || unstack(x, y)
               if on-table(x):
                   apply: on-table(x), clear(x), handsfree || pick-table(x)
clear(x)    ⤳  if holding(x): apply: holding(x) || place-table(x)
               if exists y. on(y, x):
                   apply: on(y, x), clear(y), handsfree || unstack(y, x)
```

(a) The regression rules selector for Blocks World (applicable w.r.t. any constraints).

Goal⁰: on(D, E)

$$\forall x.\ \text{on-table-g}^d(x) \land \text{holding}(x) \to place\text{-}table(x)$$
$$\forall x.\ \text{on-table-g}^d(x) \land \neg\text{holding}(x) \to \text{holding-goal}^{d+1}(x)$$
$$\forall x. \forall y.\ \text{on}^d(x, y) \land \text{clear}(y) \land \text{holding}(x) \to stack(x, y)$$

Goal¹: holding(D) ⟵
$$\forall x. \forall y.\ \text{on}^d(x, y) \land \text{clear}(y) \land \neg\text{holding}(x) \to \text{holding-goal}^{d+1}(x)$$
$$\forall x. \forall y.\ \text{on}^d(x, y) \land \neg\text{clear}(y) \to \text{clear-goal}^{d+1}(y)$$
$$\forall x. \forall y.\ \text{holding-g}^d(x) \land \text{on}(x, y) \land \text{clear}(x) \land \text{handsfree} \to unstack(x)$$
$$\forall x. \forall y.\ \text{holding-g}^d(x) \land \text{on}(x, y) \land \text{clear}(x) \land \neg\text{handsfree} \to \text{handsfree-g}^{d+1}()$$

Action: unstack(D, C)
$$\forall x. \forall y.\ \text{holding-g}^d(x) \land \text{on}(x, y) \land \neg\text{clear}(x) \to \text{clear-g}^{d+1}(x)$$
......

(b) Compilation of rule selectors into First-Order Logic formulas.

Figure 3: State-dependent regression rule selector in the Blocks World domain. For brevity, we have omitted atoms in the constraint set. All rules listed above are applicable under any constraints.

**Remark.** The compilation in Theorem 4.2 can generate finite-breadth, unbounded-depth *RelNN* circuits for more problems than the compilation in Theorem 4.1. For example, in Blocks World, the number of atoms in the goal sets in BWD is unbounded. However, since the problem is of constant SOS width, it can be compiled into a finite-breadth circuit. When the search horizon $T$ is finite, both the SOS width and $k_{\text{BWD}}$ will be finite because there are only a constant number of actions that can be applied to update the goal set / constraint set. So a problem can be compiled into a finite-depth, finite-breadth *RelNN* circuit with BWD compilation if and only if it can be compiled with S-GRS compilation, although S-GRS compilation may generate smaller circuits.

### 4.3 Compilation of S-GRS with a Regression Rule Selector

Unfortunately, both constructions in the previous section require a depth-$O(T)$ *RelNN* circuit, which is impractical for most real-world problems, as $T$ can be exponential in the number of objects in a domain. In order to compile the goal regression search algorithm into a smaller circuit, we consider scenarios where the rule used to construct the (optimal) plan can be computed without doing a full search. Here the idea is to bring in human priors or machine learning to learn a low-complexity *regression rule selector* that directly predicts which goal regression rule to use. There exists such a rule selector for some problems (e.g., Blocks World) but not others (e.g., Sokoban).

Formally, we define a state-dependent *regression rule selector* (RRS) $select(s, g, cons)$, which returns an ordered list of preconditions and a ground action $a$. Its function body can be written in the following form: $g^{cons} \lleftarrow$ if $\rho(s, cons): p_1^{cons}, p_2^{cons \cup \{p_1\}}, \cdots, p_m^{cons \cup \{p_1, \cdots, p_{m-1}\}} \| a$, which reads: in order to achieve $g$ under constraint $cons$ at state $s$, if $\rho(s, cons)$ is true, we can apply the regression rule $p_1, p_2, \cdots, p_m \| a$. Formally, $select$ is composed of a collection of rules. Each rule is a tuple of $\langle args, \rho, pre, a, g, cons \rangle$, where $args$ is a set of variables, $\rho$ is a FOL formula that can be evaluated at any given state $s$, $pre$ is an ordered list of preconditions, $a$ is a ground action, $g$ is the goal atom, and $cons$ is the constraint. $\rho, pre, a, g$, and $cons$ may contain variables in $args$. Such a rule selector can be implemented by hand (as a set of rules, as illustrated in Fig. 3a for the simple BlocksWorld domain), or learned by a *RelNN* model.

The main computational advantage is that now, for a given tuple $(s, g, cons)$, there is a single ordered list of preconditions and final action. Therefore, given the function $select$, we can simply construct a sequence of actions that achieves the goal by recursively applying the rule selector. In this section, we will first discuss scenarios where such rule selectors can be computed with shallow *RelNN* circuits.

**Circuit complexity of regression rule selectors.** We first discuss scenarios where there exist finite-depth, finite-breadth circuits for computing the regression rule selector. Note that there are three "branching" factors in choosing the regression rule: 1) the operator to use, 2) the order of the preconditions, and 3) the binding of "free" variables that are not mentioned in the goal atom.

First, in many cases, a simple condition on the state determines which operator is appropriate to use, and the preconditions can be achieved in a fixed order. Second, in many cases, the unbound variables in a rule are either determined by early preconditions or can be bound arbitrarily. When these values must be chosen very carefully (e.g.,when there are resource constraints), then a simple regression rule selector may not exist. For example, in Blocks World, to achieve *clear*(B), then the optimal way is to perform *unstack*(*A*, *B*), where A is the object on B. Furthermore, in this case, the precondition order is fixed: *clear*(B) $\leftsquigarrow$ *on*(A, B), *clear*(A), *handsfree*() $\|$ *unstack*(A, B) for arbitrary pairs of objects A, B. In the following, we consider cases where the regression rule selector can be computed with a first-order logic formula, therefore a finite-depth and finite-breadth *RelNN* circuit.

**Serializability of RRS.** Given a regression rule selector *select*, we can obtain a single trajectory (if *g* is reachable from *s* under constraints *cons*). We define $tr^{select}(s, g, cons)$ as the trajectory returned by recursive application of *select*. It returns $\perp$ if no plan can be found.

**Definition 4.1** (RRS Serializability). *A regression rule selector is serializable if and only if the following condition holds for any state s, any goal g, and any constraint set. Consider the regression rule returned by select(s, g, cons)* $\leftsquigarrow$ $p_1, p_2, \cdots, p_k \| a$. *If g is achievable from s, then it can be achieved via the concatenation of the following trajectories:* $\overline{a}_1 = tr^{regress}(s, p_1, constraint)$, $\overline{a}_2 = tr^{regress}(\mathcal{T}(s, \overline{a}_1), p_2, constraint \cup \{p_1\})$, $\cdots$, $\overline{a}_k$, *and* $\{a\}$.

**Compiling policy neural networks.** If there exists a regression rule selector that is computable with a finite-depth, finite-breadth *RelNN* circuit, it will be possible to construct a policy for the original problem with another *RelNN* circuit. This construction can be *dramatically* more efficient than the general constructions in Section 4.2. Let *k* denote the "width" of the regression rule selector, i.e., the maximum size of the constraints we need to keep track of when applying *select*.

**Theorem 4.3** (Compilation of S-GRS with a regression rule selector). *Given a regression rule selector* select *(i.e., the state-constraint condition function ρ) that can be computed by a relational neural network in RelNN[$D_r, B_r$], for any planning problem P, if* $tr^{select}(s_0, g, \emptyset) \neq \perp$, *letting d be the depth of the regression tree, then there is a RelNN circuit for the problem in RelNN[$O(d \cdot D_r), \max(B_r, \beta)$], where β is the maximum arity of predicates.*

*Proof.* For any step *i*, let $pre_{\leq i}$ be the length *i* prefix of the precondition list. We define $Goal^d$ to be the single goal atom and set of constraints we need to satisfy at depth *d*. We have the following rules:

$$\forall d. \forall i < |pre|. \left(Goal^d = (g, cons)\right) \wedge \rho(s, cons) \wedge pre_{\leq i} \wedge \neg pre_{i+1} \Rightarrow \left(Goal^{d+1} = (pre_{i+1}, cons \cup pre_{\leq i})\right)$$

$$\forall d. \left(Goal^d = (g, cons)\right) \wedge \rho(s, cons) \wedge pre \Rightarrow a$$

Here the evaluations of preconditions *pre* are based on the current state (input to the policy network) and they can be evaluated in parallel. We illustrate this construction in Fig. 3b. The first set of rules computes the next subgoal (and constraint set) to achieve, while the second set of rules outputs the next action when all preconditions of the current rule have been satisfied (which will be the output of the entire policy). Since at each layer *d*, we only need to keep track of one tuple of (*g*, *cons*), the breadth of the circuit is $\max(B_r, \beta)$, where β is required to store the input state. $\square$

This construction yields a significant reduction in breadth (from tracking all conjunctions up to $k_{BWD}$ atoms to only one tuple of (*g*, *cons*)), as well as a reduction of the number of layers for the *RelNN* policy, depending on the domain. In particular, if the average number of preconditions that need to be recursively solved in each rule is *b* (roughly equal to the number of preconditions in each rule), the depth of the regression tree is $d = O(\log_b T)$, where *T* is the planning horizon.

Finally, for some problems, even if we have a regression rule selector, to compute a plan, we may still need an unbounded depth of search steps (i.e., *d* depends on the number of objects in the domain). For example, in Blocks World, to achieve *clear*(A), the number of steps needed is proportional to the number of objects on top of A. There are two ways to build a relational neural network policy in this case. First, we can allow the network to have a variable number of iterations by applying the same layer recurrently several times. Second, there might be "shortcuts" that we can learn. We discuss both solutions in Appendix B.3. These methods allow us to construct finite-depth *RelNN* circuits for a variety of domains, including Blocks World, and path-finding in any acyclic maze.

## 5 Problem Analysis and Results

We first analyze the regression width and circuit complexities for familiar AI planning problems. Table 1 summarizes the results on a simplified setting where the goal of the planning problem is a

| Problem | Constant Breadth? | Depth |
|---|---|---|
| BlocksWorld | ✓ ($k = 1$) | Unbounded |
| Logistics | ✓ ($k = 0$) | Unbounded |
| Gripper | ✓ ($k = 0$) | Constant |
| Rover | ✓ ($k = 0$) | Unbounded |
| Elevator | ✓ ($k = 0$) | Unbounded |
| Sokoban | ✗ | Unbounded |

Table 1: Width and circuit complexity analysis for 6 problems widely used in AI planning communities. For problems with a constant circuit breadth, we also annotate their regression width $k$. "Unbounded" depth means that the depth depends on the number of objects.

| Task | Model | n=10 | n=30 | n=50 |
|---|---|---|---|---|
| As3 | *RelNN*[1, 2] | $0.2_{\pm 0.05}$ | $0.02_{\pm 0.01}$ | $0.0_{\pm 0.0}$ |
| As3 | *RelNN*[2, 2] | $\mathbf{1.0}_{\pm 0.0}$ | $\mathbf{1.0}_{\pm 0.0}$ | $\mathbf{1.0}_{\pm 0.0}$ |
| Log. | *RelNN*[3, 2] | $1.0_{\pm 0.0}$ | $0.30_{\pm 0.03}$ | $0.23_{\pm 0.05}$ |
| Log. | *RelNN*[$f(n)$, 2] | $1.0_{\pm 0.0}$ | $1.0_{\pm 0.0}$ | $1.0_{\pm 0.0}$ |
| BW | *RelNN*[3, 2] | $1.0 / 2.9_{\pm 0.4}$ | $1.0 / 10.2_{\pm 0.45}$ | $1.0 / 15.7_{\pm 2.5}$ |
| BW | *RelNN*[$f(n)$, 2] | $\mathbf{1.0 / 2.5}_{\pm 0.4}$ | $\mathbf{1.0 / 3.1}_{\pm 0.5}$ | $\mathbf{1.0 / 3.5}_{\pm 0.4}$ |

Table 2: Success rate of learned policies in different environments. For Assembly3 (As3) and Logistics (Log.), we show the success rate. For Blocks World (BW), we show the success rate / average solution length. We choose $f(n) \stackrel{\text{def}}{=} n/5+1$ for Logistics and $f(n) \stackrel{\text{def}}{=} n/10 + 3$ for BlocksWorld. In the notation of *RelNN*[$D, B$], $D$ is the number of layers and $B$ is the maximum arity of edges.

single atom. In summary, most of these problems have a constant breadth (i.e., the regression width of the corresponding problem is constant) except for Sokoban. Most problems have an unbounded depth: that is, the depth of the circuit will depend on the number of objects (e.g., the size of the graph or the number of blocks). For problems in this list, when there are multiple goals, usually the goals are not serializable (in the optimal planning case). If we only care about satisficing plans, for Logistics, Gripper, Rover, and Elevator, there exists a simple serialization for any conjunction of goal atoms (basically achieving one goal at a time). See Appendix C for more detailed proofs and additional discussions of generalization to multiple goals and suboptimal plans.

Next, given this analytical understanding of the relationship between planning problems and their policy circuit complexity, we perform some simple experiments to see whether that relationship is borne out in practice. Relational neural networks have been demonstrated to be effective in solving some planning problems [Dong et al., 2019, Jiang and Luo, 2019, Li et al., 2020], but their complexity has not been systematically explored. We consider two families of problems: one predicted to require finite depth and one predicted to require unbounded depth. For all tasks, we use Neural Logic Machines [Dong et al., 2019] as the model; we set the number of hidden neurons in the MLP layers to be sufficiently large (64 in all experiments) so that it is not the bottleneck for network expressiveness. We use the same encoding style for all problems (the graph-like relationship). Therefore, whether different problems have bounded or unbounded width primarily depends on the available goal regression rules, how these rules can be serialized, and the width of the rules. We show the average performance across 3 random seeds, with standard errors.

**Assembly3: finite depth circuits.** The domain Assembly3 contains $n$ objects. Each object has a category, chosen from the set $\{A, B, C\}$. The goal of the task is to select three objects $o_A$, $o_B$, and $o_C$ sequentially, one for each category, while satisfying a matching constraint: *match*($o_A, o_B$) and *match*($o_B, o_C$). Therefore, to select the first object (e.g., an object of type $A$), the policy needs to perform two layers of goal regression (first find the set of possible $B$-typed objects that match the object, and then find another $C$-typed object). We trained two models. Both models have breadth 2, but the first model has only 1 layer (theoretically not capable of representing the two-step goal regression), while the second model has 2 layers. Shown in Table 2, we train models on environments with 10 objects and test them on environments with 10, 30, and 50 objects. The first model is not able to learn a policy for the given task, while the second exhibits perfect generalization.

**Logistics: unbounded depth circuits.** We also construct a simple Logistics domain with only cities and trucks (no airplanes), in which the graphs are not strongly connected (by first sampling a directed tree and then adding forward-connecting edges). We train the policy network on problems with less than n=10 cities and test it on n=30 and n=50. Here, we trained two policies. The first policy has a constant depth of 3, and the second policy has a depth of $D = n/5 + 1$, which is a manually chosen function such that it is larger than the diameter of the graph. Shown in Table 2, the first model fails to generalize to larger graphs due to its limited circuit depth. Intuitively, to find a path in a graph by recursively applying the regression rule, we need a circuit of an adaptive depth (proportional to the length of the path). By contrast, the second model with adaptive depths generalizes perfectly.

**BlocksWorld-Clear: unbounded depth circuits.** Based on the goal regression analysis, in BlocksWorld, in order to achieve the goal atom *clear*(A) for a specific block $A$, we would need

a circuit that is as deep as the number of objects on *A* in the initial state. We follow the same training and evaluation setup as in Logistics. Our analysis focuses on the length of the generated plan. Shown in Table 2, although both policy networks accomplish the task with a 1.0 success rate (because there is a trivial policy that randomly drops clear blocks onto the table), the policy whose depth depends on the number of blocks successfully finds a plan that is significantly shorter than fixed-depth policy. This suggests the importance of using RelNNs with a non-constant depth for certain problems.

## 6 Related Work and Conclusion

Most of the existing work on planning complexity considers the NP-Completeness or PSPACE-Completeness of particular problems, such as the traveling salesman problem [Karp, 1972], Sokoban [Culberson, 1997], and Blocks World [Gupta and Nau, 1992]. In general, the decision problem of plan existence is PSPACE-Complete for atomic STRIPS planning problems [Bylander, 1994]. Seminal work on fine-grained planning problem complexity introduces planning width [Chen and Giménez, 2007] and the IW algorithm [Lipovetzky and Geffner, 2012, Drexler et al., 2022]. Our work is greatly inspired by these ideas and strives to connect them with circuit complexity.

The concept of serialized goal regression is not completely new. These rules are particularly related to the methods in hierarchical task networks [Erol et al., 1994] and can also be seen as special cases of hierarchical goal networks [Alford et al., 2016], derived by dropping preconditions [Sacerdoti, 1974]. In addition, others, including Korf [1987] and Barrett and Weld [1993], have characterized different degrees of serializability of subgoals, but their analysis is not as fine-grained as this.

Our circuit complexity analysis builds on existing work on relational neural network expressiveness, in particular, GNNs and their variants. Xu et al. [2019] provides an illuminating characterization of GNN expressiveness in terms of the WL graph isomorphism test. Azizian and Lelarge [2021] analyze the expressiveness of higher-order Folklore GNNs by connecting them with high-dimensional WL-tests. Barceló et al. [2020] and Luo et al. [2022] reviewed GNNs from the logical perspective and rigorously refined their logical expressiveness with respect to fragments of first-order logic. Our work extends their results, asking the question of what planning problems a *RelNN* can solve.

Our work is also related to how GNNs may generalize to larger graphs (in our case, planning problems with an arbitrary number of objects). Xu et al. [2020, 2021] have studied the notion of *algorithmic alignment* to quantify such structural generalization. Dong et al. [2019] provided empirical results showing that NLMs generalize in Blocks World and many other domains. Buffelli et al. [2022] introduced a regularization technique to improve GNNs' generalization and demonstrated its effectiveness empirically. Xu et al. [2021] also showed empirically on certain algorithmic reasoning problems (e.g., max-Degree, shortest path, and the n-body problem). In this paper, we focus on constructing policies that generalize instead of algorithms, using a similar idea of compiling specific (search) algorithms into neural networks.

**Conclusion.** To summarize, we have illustrated a connection between classical planning width, regression width, search complexity, and policy circuit complexity. We derive upper bounds for search and circuit complexities as a function of the regression width of problems and planning horizon. These results provide an explanation of the success of relational neural networks learning generalizable policies for many object-centric domains. The compilation algorithms highlight that when there are resource constraints such as free space, agents' inventory size, etc., a small policy circuit may not exist. This includes cases such as Sokoban and general task and motion planning. Furthermore, our idea of serialization can be generalized to hierarchical decomposition of problems; it can be potentially extended to other domains such as optimization and planning under uncertainty.

Although all of the analyses in this paper have been done for fully discrete domains by analyzing first-order logic formulas, understanding the implications in relational neural network learning is important because, ultimately, *RelNNs* can be integrated with perception and continuous action policy networks to solve realistic robotic manipulation problems (e.g., physical Blocks World [Li et al., 2020]). It is challenging to directly generalize our proof to scenarios where states and actions are continuous values because in these cases, there will be an infinite number of possible actions (e.g., the choice of grasping poses, trajectories, etc.). We believe that it is an interesting question to investigate similar "width"-like definitions in sampling and discretization-based continuous planning algorithms. So far, we have only analyzed the circuit complexity of policies for individual planning problems; an interesting future direction is to extend these analyses to the learning and inference of "general" policies such as large language models that can generate plans for many different planning domains.

**Acknowledgement.** We thank anonymous reviewers for their valuable comments. We gratefully acknowledge support from ONR MURI grant N00014-16-1-2007; from the Center for Brain, Minds, and Machines (CBMM, funded by NSF STC award CCF-1231216); from NSF grant 2214177; from Air Force Office of Scientific Research (AFOSR) grant FA9550-22-1-0249; from ONR MURI grant N00014-22-1-2740; from ARO grant W911NF-23-1-0034; from the MIT-IBM Watson AI Lab; from the MIT Quest for Intelligence; and from the Boston Dynamics Artificial Intelligence Institute. Any opinions, findings, and conclusions or recommendations expressed in this material are those of the authors and do not necessarily reflect the views of our sponsors.

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

# Appendix

The appendix of the paper is organized as the following. In Appendix A, we provide the proof details and more discussion with related work on search complexity. In Appendix B, we provide the proof details and more discussion on policy circuit complexity. Next, in Appendix C, we include a detailed analysis of regression widths and circuit complexities for empirical planning problems. Finally, in Appendix D, we describe environmental setups and implementation details for the experiments.

## A  Proofs and Discussions for Search Complexity

### A.1  Complete and Optimal Version of S-GRS

Algorithm 3 shows a complete and optimal version of the serialized goal regression search algorithm. The key difference between this variant and the simplified one presented in the paper is that now we keep track of multiple possible trajectories that can achieve the specified goal. The algorithm is complete because for any set of precondition atoms, in the optimal trajectory, there will always be an order in which they have been achieved (note that some of the sub-trajectories can be empty, indicating that while planning for the prefix the next subgoal has already been achieved; this covers the "parallel" precondition achievement case.). We show an example of this branching factor in Figure 4. Note that, to make sure that the algorithm eventually terminates, we need to make sure that all trajectories do not have loops.

However, this bare algorithm can be very slow: the worst time complexity of it is $O(3^N)$ with respect to $N$, the number of atoms in $\mathcal{P}_0$ — in contrast to $O(2^N)$ for a simpler backward search algorithm[‖]. This is because the possible state set for different subgoals and different constraints may overlap, so it will "waste" time during the search. Therefore, this algorithm may not be of practical use.

---

**Algorithm 3** Serialized goal regression search with multiple optimal path tracking.

---

**function** GRSOPT($s_0$, $g$, *cons*)
  **if** $g \in s_0$ **then return** ($s_0$)
  *all_possible_t* = empty_set()
  **for** $r \in \mathcal{R}_0$: *cons* $\cap$ *eff*_($action(r)$) $= \emptyset \wedge goal(r) = g$ **do**
    $p_1, p_2, \cdots, p_k$ = subgoals($r$)
    **if** $\exists p_i$ s.t. $p_i \in$ *goal stack* **then**
      **continue**
    *possible_t* = $\{\emptyset\}$
    **for** $i$ in $1, 2, \cdots, k$ **do**
      *next_possible_t* = empty_set()
      **for** *prefix_t* in *possible_t* **do**
        *imm_state* = last_state(*prefix_t*)
        **for** $nt \in$ GRSOPT(*imm_state*, $p_i$, *cons* $\cup \{p_1, \cdots, p_{i-1}\}$) **do**
          *next_possible_t* = *next_possible_t* $\cup$ {*prefix_t* + *nt*}
      *possible_t* = *next_possible_t*
    *all_possible_t* = *all_possible_t* $\cup$ $\{t + \mathcal{T}$ (last_state($t$), $action(r)) \mid t \in$ *possible_t*$\}$
  **return** *all_possible_t*

---

### A.2  Proof for Theorem 3.1: S-GRS Optimality and Completeness

We define the set of optimally serializable rules for a state $s$ as $OSR(s)$, and the set of single-literal goals that can be solved with $OSR(\cdot)$ from $s_0$ as $OSG(s_0)$.

*Proof.* Completeness: the definition of $OSG(s_0)$ asserts that $g$ can be solved by S-GRS.

Optimality: proof by induction. In the following, we prove that for any $g \in OSG(s_0)$, the trajectory returned by S-GRS is optimal.

---
[‖]Note that *constraints* is always a subset of state *state*, so the worst complexity is $O(3^N)$.

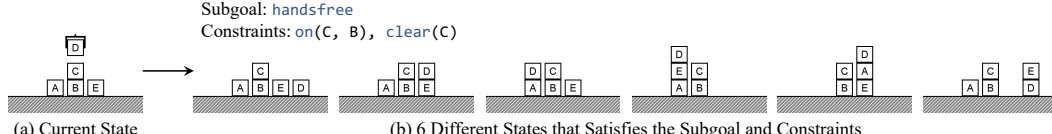

(a) Current State              (b) 6 Different States that Satisfies the Subgoal and Constraints

Figure 4: Illustration of the branching factor caused by tracking multiple resulting states after achieving a subgoal.

First, we consider the set of all S-GRS calls in the regression tree that contributes to the returned trajectory. We do induction based on a distance metric. Consider any goal atoms $g \in OSG(s_0)$, let $Distance(s_0, g, cons)$ be the length of the path returned by S-GRS. First, for all argument tuples with distance 0 (i.e., $g \in s_0$ and $cons \subseteq s_0$), S-GRS will return the optimal trajectory.

Next, assuming that S-GRS will return optimal trajectories for all argument tuples with distances strictly smaller than $d$. Consider a new argument call $grs(s_0, goal, cons)$ with distance $d$. Consider now all optimally serializable and goal-achieving rules $r \in OSR(s_0)$ such that $goal(r) = g$ and $cons \cap eff_-(action(r)) = \emptyset$. Let $r_g$ be the rule that returns the plan during search, and $p_1 \cdots p_k$ be the serialized precondition list. First, all sub-function calls to S-GRS will return optimal plans by induction. Second, by optimal serializability of the rule $r$, we know that concatenating the sub-plans will yield an optimal plan to achieve $g$ while preserving $g$. Therefore, our induction holds. □

## A.3 Making Non-SOS Problem Serializable

Here we discuss a general strategy to make non-SOS problems serializable by introducing super predicates. In particular, for a given operator, if two of the preconditions (without loss of generality, say, $p(x)$ and $q(x)$) are not serializable, we can introduce a new predicate $p\_and\_q(x)$, and rewrite the precondition of the operator with this new super predicate.

The disadvantage of this encoding is that it will enlarge not only the predicate set but also the operator set. In particular, for any operator $o$ such that $p(x) \in eff_+(o)$ or $p(x) \in eff_-(o)$, we need to break the operator into two sub-operators. For example, if $p(x) \in eff_+(o)$, we need to create a new operator $o_1$ that has $\neg q(x)$ in its precondition, and set $eff_+(o_1) = eff_+(o)$ and $eff_-(o_1) = eff_+(o)$. For the second operator $o_2$, it should have $q(x)$ in its precondition and $eff_+(o_1) = eff_+(o) \cup \{p\_and\_q(x)\}$ and $eff_-(o_1) = eff_-(o)$. As the number of predicates in this super predicate increases, the number of additional operators grows exponentially.

This provides a characterization of "what contributes to the (forward) width of a problem." Specifically, we have identified two sources of forward widths: non-serializability of certain operators, and the need to track constraints while doing a regression search.

## A.4 Proof for Theorem 3.2: Polynomial Hardness for SOS-k problems.

If a problem $P$ is of SOS width $k$, it can be solved in time $O(N^{k+1})$, where $N$ is the number of atoms. Here we have omitted polynomials related to enumerating all possible actions and their serializations (which are polynomial w.r.t. the number of objects in $\mathcal{U}$).

*Proof.* First, if we are given the rule set $\mathcal{R}$, then the proof can be simply done by analyzing function calls: it is important to observe that there are at most $O(N^{k+1})$ possible argument combinations of $(goal, cons)$ to function GRS.

First, due to SOS condition, we only need to maintain one single optimal trajectory for each set of argument values $(s_0, goal, cons)$. Next, we consider we only maintain one single optimal trajectory for each pair of $(goal, cons)$, ignoring the first argument $s_0$. Note that, even if the "cached" trajectory does not include the input state $s_0$ we can still return the cached trajectory. This is because the optimal serializability suggests that any trajectories for achieving $(goal, cons)$ can be extended to future preconditions.

Finally, we consider removing the assumption of knowing $\mathcal{R}$. Note that the enumeration of all possible goal width-$k$ regression rules can be done in $O(2^k)$, which is considered constant when $k$ is small.

Therefore, overall the algorithm runs in polynomial time and the solver does not need to know the set of optimally serializable regression rules $\mathcal{R}$ a priori. □

First, the fact that the algorithm does not need to know $\mathcal{R}$ *a priori* is important because it suggests that, as long as there exists a set of optimally serializable rules for the problem we are trying to solve, the search algorithm will find a solution, without having any additional knowledge other than the operator definitions. Intuitively, this is because doing S-GRS with additional non-optimally serializable rules (e.g., using all rules in $\mathcal{R}_0$) will not falsify the completeness and optimality of the algorithm.

Second, unfortunately, the SOS condition can not be relaxed. That is, if we only assume optimal serializability of generalized regression rules and all rules have width $k$, the proof will fail. This is because even if regression rules have a small width, there will be a possibly exponential branching factor caused by "free variables" in regression rules that do not appear in the goal atom. Concretely, let us consider the following rule $g(x) \leftsquigarrow p_1(x, y)^\emptyset, p_2(x)^\emptyset \parallel a_1(x, y)$. This rule is of width 0; however, since there is no guarantee that any optimal trajectory for achieving $p_2(x)$ will also achieve $p_1(x, y) \wedge p_2(x)$, during search, we can not use the cached trajectory for $(p_2(x), \emptyset)$ as the return value when we solve for $p_2(x)$ after achieving $p_1(x, y)$, which breaks the complexity analysis.

### A.5 Additional Discussion

**Deriving optimally-serializable, small-width rules.** For a practical system, if we can know the set of regression rules needed, we can accelerate the search significantly. Here we discuss a strategy for deriving such rules by analyzing the domain definition itself.

If we ignore the optimality, a sufficient condition that a rule $r$ is not strong optimal serializability for a given state $s_0$ is that, there exists a precondition $p_i$ such that there is no rule $r'$ such that $goal(r') = p_i$, $eff_-(action(r')) \cap p_i = \emptyset$ and $goal(r) \notin subgoals(r')$. The last condition is the goal-stack checking: if achieving a goal $g$ using rule $r$ requires achieving goal $g$ itself, then the rule is not serializable.

Putting this into practice, consider all rules for achieving $clear(A)$. If there is another object B on A in $s_0$, then only the rule derived from $unstack(B, A)$ will be SOS serializable. This is because, for any other rules $unstack(y, A)$, where $y \neq B$, achieving $on(y, A)$ will involve achieving $clear(A)$ itself. Note that this analysis can be done by static (lifted) analysis of the domain.

This analysis can be viewed as simulating a one-step goal regression search in a lifted way, and it can be extended to more steps. In some domains, this strategy will help rule out some serialization of the preconditions (such as the BlocksWorld example), but in general, whether a rule is strong optimally serializable is state-dependent and can be hard to compute.

**Connection with other concepts in search.** Another view of the regression width is the maximum number of constraints one needs to track while applying S-GRS recursively. This notion is analogous to the treewidth of a constraint graph in constraint satisfiability problems (CSPs): intuitively, the number of backward-pointing edges [Freuder, 1982].

As also briefly discussed in Lipovetzky and Geffner [2012], delete-relaxation heuristics are also related. In particular, the hFF heuristic can be interpreted as applying backward search by considering "parallelizability" in contrast to "serializability." Informally, for a given state $s$ and a sequence of preconditions $p_i$, we compute trajectories for each individual $p_i$ separately and concatenate them.

**Generalization to $\forall$-quantified preconditions.** It would be possible to extend the definition to $\forall$-quantified preconditions by simply allowing regression rules to have an unbounded number of subgoals (e.g., to handle goals such as $\forall x on\text{-}table(x)$). However, this is not helpful in general because the constraints will accumulate as the number of subgoals becomes larger, and enumerating subgoal serialization will now take $O(m!)$ where $m$ is the number of subgoals. This notion will be more helpful when we have a "goal regression rule selector."

### A.6 Connection with Width-Based Forward Search IW(k)

Wecall the width of a problem defined in Chen and Giménez [2007] and the Iterated Width algorithm $IW(k)$ [Lipovetzky and Geffner, 2012] as forward width. Then we have:

**Theorem A.1** (Sufficiency condition for forward width). *Any planning problem that has SOS-width k has a forward width of at most k + 1 and, hence, can be solved by the algorithm IW(k + 1).*

*Proof.* We prove that for any trajectory returned by the S-GRS algorithm for an SOS width-$k$ problem $P$, all actions in the trajectory achieve a novel $k + 1$-sized atom set. If this is true, then the final goal will be reachable from $s_0$ in the size $k + 1$ tuple graph in IW search.

Consider any action $a$ in the returned plan, it corresponds to a unique rule for achieving a subgoal $g$ and a set of constraints *cons*. If the size $k + 1$ tuple $\{g\} \cup cons$ has already been achieved before $a$, then due to the SOS condition, we can make the returned plan shorter. This violates the optimality of the S-GRS algorithm. $\square$

Unfortunately, these two definitions are not exactly equivalent to each other. For example, consider the Blocks World domain and a state of three stacked blocks A, B and C, the goal regression rule

$$clear(A) \leftsquigarrow on(B, A), clear(B)^{\emptyset}, handsfree()^{\{clear(B)\}} \parallel unstack(y, x)$$

is SOS. It has width 1 because while achieving *handsfree*(), we have to explicitly maintain *clear*(B); otherwise, it is possible that we directly put $C$ back onto $B$ instead of putting $C$ onto the table (both plans are optimal for achieving *handsfree*()) after achieving *clear*(B). By contrast, in IW, the Blocks World domain has width 1 instead of 2 if the goal predicate is *clear*, because the optimal plan for achieving $handsfree()^{\{clear(B)\}}$ happens to be the optimal plan for achieving *on-table*(C).

Proofs for (forward) widths of planning problems in IW were mostly done by analyzing solution structures. By contrast, the constructions in this paper form a new perspective to interpret planning problem width: they are subgoals and constraints that need to be satisfied during a goal regression search. This view is helpful because our analyses focus on analyzing concrete goal regression rules, and the derived results are compositional. Unfortunately, not all problems that can be solved by $IW(k)$ have a regression width of $k - 1$.

# B    Proofs and Discussions for Policy Circuit Complexity

## B.1    Proof of Theorem 4.1: Compilation of BWD

Given a planning problem $P$, let T be the length of the optimal trajectory as (the planning horizon), $k_{\text{BWD}}$ be the maximum number of atoms in the goal set in BWD, and $\beta$ be the maximum arity of atoms in the domain. The backward search algorithm BWD can be compiled into a relational neural network in $RelNN[O(T), \beta \cdot k_{\text{BWD}}]$.

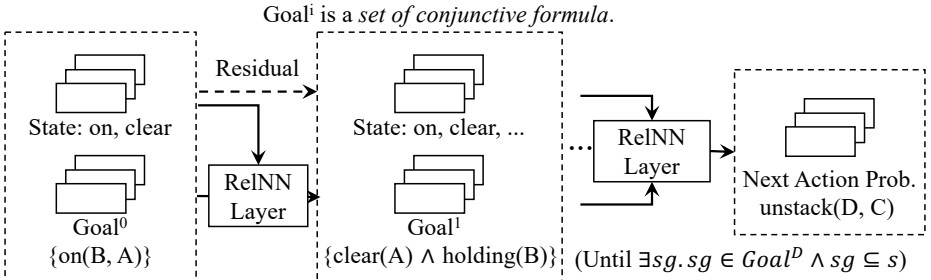

Figure 5: Illustration of the compilation of backward search into a *RelNN* policy.

*Proof.* There is a proof by construction. As illustrated in Fig. 5, a relational neural network can use its intermediate representations to keep track of a set of subgoals (i.e., a set of sets of ground atoms) $Goal^d$ at each layer $d$. Initially, $Goal^0$ contains only the goal atom. Then, $sg \in Goal^d$ if there is a path of length $d$ from any state $s$ that satisfies $sg \subseteq s$ to the final goal of the planning problem. For all layers $d$, for any subgoals $sg$, for all ground actions $a \in \mathcal{A}$, $sg \in Goal^d \wedge (eff_-(a) \cap sg = \emptyset) \Rightarrow (sg \cup pre(a) \setminus eff_+(a)) \in Goal^{d+1}$. Since the maximum number of atoms in BWD is $k_{\text{BWD}}$, we only need to keep track of ground atom conjunctions that involve at most $k_{\text{BWD}} \cdot \beta$ distinct objects. Therefore, BWD search can be compiled into a relational neural network in $RelNN[O(T), k_{\text{BWD}} \cdot \beta]$.

A more detailed construction is the following. Let $k' = k_{\text{BWD}} \cdot \beta$. Let $x_1, \cdots, x_{k'}$ be variables. $\{q_i(x_1, \cdots, x_{k'})\}$ is the set all possible conjunctive expressions that involve variables $\{x_1, \cdots, x_{k'}\}$. Let $q_i^{(d)}(x_1, \cdots, x_{k'})$ be the value of $q_i(x_1, \cdots, x_{k'})$ at layer $d$, and $expr(q_i)$ be the conjunctive expression of $q_i$. For all $d$, $q_i$, $q_j$, and for all actions $a$ that are grounded on variables from $\{x_1, \cdots, x_{k'}\}$, if $(eff_-(a) \cap expr(q_i) = \emptyset) \wedge (expr(q_j) = expr(q_i) \cup pre(a) \setminus eff_+(a))$, there is a logical rule: $\forall x_1, \cdots, x_{k'}. \, q_i^{(d)}(x_1, \cdots, x_{k'}) \Rightarrow q_j^{(d+1)}(x_1, \cdots, x_{k'})$. Furthermore, for all $d$, $q_i$, if $(eff_-(a) \cap expr(q_i) = \emptyset) \wedge (expr(q_i) \cup pre(a) \setminus eff_+(a) \subset s)$, where $s$ is the current state, we have the following rule that outputs the next action to take, $\forall x_1, \cdots, x_{k'}. \, q_i^{(d)}(x_1, \cdots, x_{k'}) \Rightarrow a^{(d+1)}$, where $a^{(d)} = 1$ indicates that there exists a length $d$ trajectory for achieving the goal. In practice, we will the shortest one. Note that all these rules are fully quantified by $\forall x_1, \cdots, x_{k'}$ so they are lifted FOL rules and can be implemented by a *RelNN* circuit. $\square$

## B.2    Proof of Theorem 4.2: Compilation of S-GRS

Given a planning problem $P$ of SOS width $k$, let T be the length of the optimal trajectory, and $\beta$ be the maximum arity of atoms in the domain. The serialized goal regression search S-GRS can be compiled into a $RelNN[O(T), (k + 1) \cdot \beta]$, where $k + 1 \leq k_{\text{BWD}}$.

*Proof.* There is a proof by construction. The easiest construction is based on the sufficient condition between regression width $k$ and $IW(k)$. In particular, based on Theorem 3.3, we know that $IW(k + 1)$ can solve the planning problem $P$. Therefore, we can use the following construction to simulate $IW(k + 1)$.

The most important trick in the construction is that a relational neural network can use its intermediate representations to keep track of two sets of atoms. First, for each atom tuple $t$ up to size $k + 1$, whether $t$ is reachable from $s_0$ within $d$ steps. Second, let $s_t$ be the last state of the optimal trajectory associated with $t$; for each atom tuple $t$ up to size $k + 1$ and for a predicate $p(x_1, x_2, \cdots, x_\beta)$, we

keep track of whether $p(x_1, x_2, \cdots, x_\beta)$ is true in $s_t$. At each layer $d$, we consider all possible ground actions $a \in \mathcal{A}$, if there exists such $(t_1, t_2)$ such that $t_2$ is not reachable within $d$ steps and $t \subseteq \mathcal{T}(s_{t_1}, a)$, we set $t_2$ to be reachable within $d + 1$ and $s_{t_2} = \mathcal{T}(s_{t_1}, a)$.

In addition to the realizability of the $IW(k + 1)$ algorithm, we need to additionally prove that, if we follow the policy derived by simulating $IW(k + 1)$ for the first step, we can recurrently apply this policy. To see that, consider the path from $s_0$ to the goal state in $IW(k + 1)$ in the tuple graph. Since we are moving forward by one step in this tuple path, doing a search from the second tuple in the path (i.e., applying the same *RelNN* policy at the second world state) will yield the same trajectory.

Similar to the construction in Theorem 4.1, all the rules are lifted FOL rules and can be implemented by a *RelNN* circuit. It is also possible to use the same trick to compile the original S-GRS algorithm: essentially, for each $(g, cons)$ pair, keep track of the optimal trajectory for achieving it. However, the construction will be more complicated and is omitted here. □

## B.3 More Discussion

For some problems, even if we have a regression rule selector, to compute a plan, we may still need an unbounded depth of search steps (i.e., $d$ depends on the number of objects in the domain). For example, in Blocks World, to achieve *clear*(A), the number of steps needed is proportional to the number of objects on top of A. There are two ways to build a relational neural network policy in this case. First, we can allow the network to have a variable number of iterations by applying the same layer recurrently several times. We have already illustrated this in the experiment.

Second, there might be "shortcuts" that we can learn. An intuitive example is that in BlocksWorld, the goal regression for *clear*(A) will recur until we have found the topmost block stacked on top of A. If the domain has a new predicate *above*(x, y) indicating whether the block $x$ is above another block $y$, then we can use just one single rule to predict the topmost block on top of A: the topmost block B on A satisfies *above*(B, A) ∧ *clear*(B).

The same technique also applies to the problem of path-finding in any acyclic maze. Recall that a classical solution to any acyclic maze is that we only make right turns. This can be cast as a linearization of the entire maze by doing a depth-first search from the start or the goal location. In particular, starting from the goal location if we only do right turns, we will get a sequence of the edges of the maze such that the start edge is one of the elements (if there is a path from the start to the goal). Therefore, doing goal regression under this linearized maze is trivial: the entire "regression tree" is just a chain of edges and there is a simple shortcut for predicting the next move.

# C  Analysis of AI Planning Problems

Here, we list six popular problems that have been widely used in AI planning communities and discuss their circuit depth and breadth. We summarize our results in the Table 3.

| Problem | Breadth | Regression Width | Depth |
|---|---|---|---|
| BlocksWorld | Constant | 1 | Unbounded |
| Logistics | Constant | 0 | Unbounded |
| Gripper | Constant | 0 | Constant |
| Rover | Constant | 0 | Unbounded |
| Elevator | Constant | 0 | Unbounded |
| Sokoban | Unbounded | N/A (not serializable) | Unbounded |

Table 3: Width and circuit complexity analysis for 6 problems widely used in AI planning communities. This table contains the same content as Table 1; we include it here for easy reference.

Note that here, we are limiting our discussion to the case where the goal of the planning problem is a single atom. In summary, most of these problems have a constant breadth (i.e., the regression width of the corresponding problem is constant) except for Sokoban. Most problems have an unbounded depth: that is, the depth of the circuit will depend on the number of objects (e.g., the size of the graph or the number of blocks. The problem gripper has a constant depth under a single-atom goal because it assumes the agent can move directly between any two rooms; therefore, no pathfinding is required. Sokoban has an unbounded breadth even for single-atom goals because if there are multiple boxes blocking the way to a designated position, the order to move the boxes can not be determined by simple rules. The constant breadth results can be proved by construction: list all regression rules in the domain. In the following Appendix C.1, we first discuss the SOS width of two representative problems, Blocks World and Logistics. All other problems can be analyzed in a similar way.

Before we delve into concrete complexity proofs, we would like to add the note that for problems in this list, when there are multiple goals, usually the goals are not serializable (in the optimal planning case). This can be possibly addressed by introducing "super predicates" that combine two literals. For example, if two goal atoms $p(x)$ and $q(x)$ are not serializable, we can introduce a new predicate $p\_and\_q(x)$, and rewrite all operators with this new super predicate. Appendix A.3 provides more discussion and examples. This will make the problem serializable but at the cost of significantly enlarging the set of predicates (exponentially with respect to the number of goal atoms).

If we only care about satisficing plans, for Logistics, Gripper, Rover, and Elevator, there exists a simple serialization for any conjunction of goal atoms (basically achieving one goal at a time).

These theoretical analyses can be used in determining the size of the policy circuit needed for each problem. For example, one should use a RelNN of breadth $(k + 1) \cdot \beta$, where $k$ is the regression width and $\beta$ is the max arity of predicates. For problems have unbounded depth, usually the depth of the circuit grows in $O(N)$, where $N$ is the number of objects in the environment (e.g., in Elevator, the number of floors).

## C.1 SOS Width Analysis

To prove that a problem can be solved optimally using S-GRS, we only need to show that there exists a goal regression rule selector. Therefore, we will be using the syntax from Section 4 to write down the set of goal regression rule selectors.

**Finite SOS width for Blocks World.**

$$\forall cons.\forall x.\forall y.on(y, x) \in s_0 \quad clear(x)^{cons} \leftsquigarrow on(y, x), clear(y), handsfree() \parallel unstack(y, x)$$
$$\forall cons.\forall x.holding(x) \in s_0 \quad clear(x)^{cons} \leftsquigarrow holding(x) \parallel place\text{-}table(x)$$
$$\forall cons.\forall x.\forall y.on(x, y) \in s_0 \quad hoding(x)^{cons} \leftsquigarrow on(x, y), clear(x), handsfree() \parallel unstack(x, y)$$
$$\forall cons.\forall x.on\text{-}table(x) \in s_0 \quad hoding(x)^{cons} \leftsquigarrow on\text{-}table(x), clear(x) \parallel pick\text{-}table(x)$$
$$\forall cons.\forall x. \quad on\text{-}table(x)^{cons} \leftsquigarrow holding(x) \parallel place\text{-}table(x)$$
$$\forall cons.\forall x.\forall y. \quad on(x, y)^{cons} \leftsquigarrow clear(y), holding(x) \parallel stack(x, y)$$
$$\forall cons.\forall x.holding(x) \in s_0 \quad handsfree()^{cons} \leftsquigarrow holding(x) \parallel place\text{-}table(x)$$

**Finite SOS width for Logistics.**

$$\forall cons.\forall v.\forall o.\forall \ell. \quad at(o, \ell)^{cons} \leftsquigarrow in(o, v), at(v, \ell) \parallel unload(y, x)$$
$$\forall cons.\forall v.\forall o.\forall \ell. \quad in(o, v)^{cons} \leftsquigarrow at(o, \ell), at(v, \ell) \parallel load(y, x)$$
$$\forall cons.\forall v.\forall \ell_1.\forall \ell_2.\forall c. \quad at(v, \ell_2)^{cons} \leftsquigarrow loc(\ell_1, c), loc(\ell_2, c), at(o, \ell_1) \parallel drive(v, \ell_1, \ell_2, c)$$
$$\forall cons.\forall v.\forall a_1.\forall a_2. \quad at(v, a_2)^{cons} \leftsquigarrow at(v, a_1) \parallel fly(v, a_1, a_2)$$

Here, we have omitted static properties for type checking (such as $a_1$ and $a_2$ need to be airports).

# D   Experimental Details

## D.1   Training details

**Environment setup.**   For the Assembly3 task, we first uniformly sample the set of type-A, type-B, and type-C object. Then, we randomly add *match* relationships so that there will be exactly only one tuple of $(a, b, c)$ that satisfies the matching condition.

For the BlocksWorld-Clear task, we use the same random sampler for initial configurations as Neural Logic Machines [Dong et al., 2019]. It iteratively adds blocks to an empty initial state. At each step, it randomly selects a clear object (or the table), and places a new object on top of it. After generating the initial state, we randomly sample an object that is not clear in the initial state as the target block.

For the Logistics task, we first randomly generate a tree rooted at the start node. This is done by iteratively attaching new nodes to a leaf node. Next, we randomly add a small number of edges to the tree. Finally, we randomly pick a node as the target node. Since we are only a small number of additional edges added to the tree, with high probability, the starting node and the end node are not in the same strongly connected component.

**Additional hyperparameters.**   The breadth, width, and hidden dimension parameters for neural networks have been specified in the experiment section and they vary across tasks. For all experiments, we use the Adam optimizer to train the neural network, with a learning rate of $\epsilon = 0.001$. Additionally, we set $\beta_1 = 0.9$ and $\beta_2 = 0.999$.

**Compute.**   We trained with 1 NVIDIA Titan RTX per experiment for all datasets, from an internal cluster.

**Code.**   We release the code for reproducing all our experimental results in https://github.com/concepts-ai/goal-regression-width.

