# OpenReview forum: "What Planning Problems Can A Relational Neural Network Solve?"
_NeurIPS.cc/2023/Conference — NeurIPS 2023 spotlight_

### Official Review · Reviewer_TK6j · 2023-06-30

**Soundness:** 3 good
**Presentation:** 2 fair
**Contribution:** 3 good
**Rating:** 7
**Confidence:** 3

**Summary:**


This paper focuses on formalizing what class of planning problems can be solved by a relational neural networks. It aims to bridge the gap between the expressivity of relational neural networks and the complexity of planning problems. To do this, a serialized goal regression rule and search algorithm is defined. Three classes of planning problems are defined based on the serialized goal regression search and show these classes can provide network parameters for relational neural network. Experiments with two planning domains show the empirical implications of the results.

**Strengths:**


* The paper provides a novel approach to quantifying the complexity of planning problems
* As far as I am aware, this is the first paper to analyze what kind of planning problems can be solved by neural networks.
* The formal analysis can aid in better evaluations of the neural approaches proposed for planning problems.


**Weaknesses:**


* Some of the notations are not clear (See questions for details).


**Questions:**

**Major**


1. Algorithm 1
    1. Plain backward search should satisfy two conditions for selection of action: $g \cup eff_{+}(a) \neq  \emptyset$ and $g \cup eff_{-}(a) = \emptyset$. The second condition is captured in the Algorithm. But the first is not.
   1. Action should be mentioned as input in Algorithm 1.
   1. `goal stack` is not defined
1. Algorithm 2
    1.  $R_o$ should be part of the input
    1.  `goal stack` is not defined
    1.  `goal_set` is not defined
    1.  Is $\pi$ a plan? If so, $\pi[-1]$ is the last action. So is $s_i$ an intermittent state or action?
    1.  If any of the $p_i$ is available in the goal stack, then that branch is not explored in the backward search? What is the benefit of doing that?
1. Line 107 mentions that there will be a goal regression rule for any possible permutation of pre(a), but as explained in Line 100 all of these are not feasible. Will the regression set contain infeasible rules as well?
1. How are the generalized goal regression rules identified? For example, consider the rule in Line 139. Can such a rule be identified automatically?
1. Were the serialized goal regression rules used in defining the relational neural network? Can explicit knowledge of such rules enable efficient learning?
1. Which relational network was used in the experiments? NLM, ReNN, or NLRL?
1. This is not a question. Just a discussion point. The traditional relational learning approaches learn rules one predicate at a time. The serialized goal-regression defines a class of planning problems that can be solved by satisfying one predicate at a time. So can this aid in understanding what class of planning problems can be solved by traditional relational learning approaches?
1. Can authors provide the classes for common IPC domains like Logistics, Gripper, Rover, Elevator, etc?

**Minor**

9. Regression rule set notation has discrepancies. In Algorithm 2, $R_0$ is used, whereas in Line 109 $R^0$ is defined.
9. There is a discrepancy in the function name. Line 111 uses S-GRS but Algorithm 2 uses GRS.
9. Is $P_0$ defined in Line 61 the same as $P_{s_0}$? The set of predicates in the initial state does not define the set of all atoms.
9. In Definition 3.1, is $\bar{a}$ a trajectory or action? For consistency, I recommend using the symbol $a$ for actions alone.
9. The term 'generalized goal regression rule' is not formally introduced. In line 141 such a rule is referred to as the reduced goal regression rule. For consistency, I recommend sticking to a single term.
9. I know the space is limited, but I would still recommend moving the connection to plan width from appendix to main paper.

**Limitations:**

---

> ### Author Rebuttal · Authors · 2023-08-08
>
> Thank you for your helpful comments!
>
> **Q1:** Notations in algorithm 1 and 2.
>
> **A1:** Thank you for all the suggestions. We will update our notations. For algorithm 1, we will include $\mathcal{A}$ as input, define `goal_set` as the set of goals to achieve during the current backward search process (initialized as the goal of the planning problem), and define the goal stack as the set of `goal_set` arguments currently in the depth-first search stack. We intentionally omitted the condition of $\textit{eff}_+$ in the algorithm to save space, because this does not affect the correctness of the algorithm (but only efficiency). We will similarly update algorithm 2 to define the goal stack and include $R$ as the input.
>
> **Q2:** Definition of $\pi$.
>
> **A2:** Thanks for the catching that! $pi$ was intended to include only a sequence of actions. Therefore, we will rewrite the line of $s_i = \pi_i[-1]$ to be $s_i = \mathcal{T}(s_{i-1}, \pi_i)$ (i.e., simulate all actions in $\pi_i$ based on $s_{i-1}$).
>
> **Q3:** Not consider $p_i$ if $p_i$ is already in the goal stack.
>
> **A3:** In short, this design is to avoid a potential infinite loop.
>
> First, adding this line will not hurt the accuracy or optimality of the search algorithm, because the search algorithm seeks  optimal trajectories to achieve a certain goal. If the subgoal in the current regression rule is a subgoal that we wish to achieve at an earlier search depth, then using this regression rule will not yield an optimal trajectory for the earlier search depth subgoal.
>
> If we do not add this condition, then there will be infinite loops: consider two regression rules for two predicates $p$ and $q$: `p <- q || a_1`, `q <- p || a_2`.
>
> **Q4:** Infeasible rules in the regression set.
>
> **A4:** Yes, the search algorithm does allow infeasible regression rules in $R$. This is because if a goal regression rule is infeasible (e.g., after achieving one of the precondition, we can not extend the current trajectory to achieve another precondition), the set `possible_t` will just be empty. Since the search algorithm enumerates all candidate rules to use, the search algorithm will return an answer as long as there exists a subset of feasible rules in the given rule set that can yield a successful plan.
>
> **Q5:** Identifying generalized goal regression rules.
> **A5:** In this paper, we did not discuss any particular algorithms to identify generalized goal regression rules. In general, in order to prove the correctness of a generalized goal regression rule, we need to verify that Definition 3.2 holds for the generalized rule. This is generally hard, and we have done this in a case-by-case manner (e.g., proving a particular generalized rule is applicable in BlocksWorld).
>
> **Q6:** Were the serialized goal regression rules used in defining the relational neural network? Can explicit knowledge of such rules enable efficient learning?
>
> **A6:** Thank you for this suggestion! Currently, the goal regression rules are not used in defining the network (we use regression rules to reason about the network size only). We think there are at least two interesting directions given your suggested idea. One would be to explicitly compile some set of given serialized goal regression rules into a network, either by manipulating network weights, or by introducing additional supervision. Another possible direction might be one where we know the basic form of the serialized regression rules, but don't know the value of some of the conditions: in such a case, we could encode those rules in a network leaving the unknown conditions "free", in the sense of representing them as an MLP, and then backprop to train just the weights in those MLPs, which we would expect to be substantially more sample efficient than learning the whole policy.
>
> **Q7:** Which relational network was used in the experiments? NLM, ReNN, or NLRL?
>
> **A7:** We were using NLMs for all the experiments.
>
> **Q8:** Connection to relational learning approaches.
>
> **A8:** Thanks for the suggestion! We think this is a very interesting idea. We think there are two possible ideas in leveraging tranditional relational learning in learning and planning. First, we can learn a PDDL-like domain definition, using traditional relational learning approaches; then, we can find ways to compile the learned definitions into a RelNN-like policy. Alternatively, one can consider learning a policy (such as all the regression rules in a domain). The difficulty with this approach is that in most domains, learning a single rule usually does not allow you to immediately solve novel problems. By contrast, multiple rules must be learned and used together in order to compute a plan. In this case, even though each regression rule only achieves only one single atom, we do not have direct supervision for them (but only learning signals when a set of rules have been learned).
>
> **Q9:** Classes for IPC planning problems.
>
> **A9:** Thanks for the great suggestion! We have included more analysis of the classic problems you suggested. Please refer to our general response for a detailed discussion.
>
> **Q10:** Notation discrepancies.
>
> **A10:** Thank you for catching these! We will fix the notation discrepancies for $R_0$, function names, and the definition for trajectories.
>
> **Q11:** Other clarifications and structures.
>
> **A11:** We appreciate all the suggestions. We will clarify that the set of predicates in the initial state does not define the set of all atoms, define generalized goal regression rule before Definition 3.2, and move back connections to plan width from appendix to the main paper.
>
> [1] Richard Li, Allan Jabri, Trevor Darrell, Pulkit Agrawal. Towards Practical Multi-Object Manipulation using Relational Reinforcement Learning.

---

> > ### Comment · Reviewer_TK6j · 2023-08-16
> >
> > Thank you for the response. I have read the rebuttal and would like to keep my score the same. Good job.

---

### Official Review · Reviewer_7eTK · 2023-07-04

**Soundness:** 3 good
**Presentation:** 3 good
**Contribution:** 2 fair
**Rating:** 6
**Confidence:** 3

**Summary:**

The paper explores the application of relational neural networks, seen as circuits, in representing policies for discrete planning problems. Specifically, it introduces a circuit complexity analysis that categorizes planning problems into three classes based on how circuit width and depth grow, by establishing connections with serialized goal regression search (S-GRS), which is a method for generating plans by working backwards from goals. In essence, the main contribution of the paper is that it combines previously introduced planning problem complexity related quantifications (width, depth) with circuit/policy complexity analysis to analyze the resource/representation requirements of policies. This analysis helps to better understand the capabilities of relational neural networks in solving planning problems, which in turn helps designing them for policy learning. By analyzing goal-conditioned policies for such planning problems, upper bounds on the circuit complexity of such policies are provided as a function of the problem's regression width. The experimental results support the theoretical formulations and demonstrate the practical applicability of the proposed approach for relatively small sized/simpler discrete planning problems.

**Strengths:**

- Originality: The paper proposes a new perspective for circuit complexity analysis for relational neural networks (GNNs, transformers), and thus what they can compute (i.e., their limits).

- Quality: The proposed method is supported by theoretical analysis and proofs. A small set of experiments were conducted to validate the approach.

- Clarity: The order of presentation is mostly nice, even though the terminology used throughout the paper sometimes makes it hard to follow.

- Significance: Helpful for constructing relational neural networks for planning problems, and the proposed formulation can also help investigate solutions to planning problems for future works in this area. If the problems analyzed had a broader scope, it would have increased the overall impact of the paper.


**Weaknesses:**

- Lack of a convincing set of quantitative results: Proposed methods might be tested on different networks and problems (only two different problems and networks are presented).

- A broader discussion of the applicability of the proposed analysis and formulation on different planning problems, especially to real-world (e.g., robotics) scenarios, would help convince the reader about the impact of the paper.

- Related work section might be placed earlier (e.g., after introduction) for better placing this work w.r.t. the literature.

- Supplementary material helps clarify some points, due to page limit it is hard to integrate this information but for clarity there might be more reference to it within the main text.


**Questions:**

- What are the main bottlenecks/issues to apply this formulation for continuous planning problems, e.g., such as the ones in robotics?

- Would this analysis and construction be still useful/practical for larger/more complex problems, which require larger networks? If yes, how?

- The problems investigated are solvable by classical search/planning methods, then what’s the advantage of using a RelNN that was constructed based on the provided analysis?

- Serialization of goal regression rules seem to be highly related to mathematical optimization / dynamical programming. Can you elaborate on this connection?


**Limitations:**

- Limitations of the work have been barely touched upon on the paper.

- The discussion on applying this formulation to continuous planning problems, e.g., in robotics, can be extended.

- In general, an evaluation and discussion on the type of problems that the proposed formulation is not applicable to would be nice.

---

> ### Author Rebuttal · Authors · 2023-08-08
>
> Thank you for your helpful comments!
>
> **Q1:** Quantitative results:
>
> **A1:** Our main objective in this paper is to formally clarify the nuances of when relational neural network circuits can realize a policy for specific planning problems, and to understand the size of the circuits necessary to achieve this. We begin by exploring discrete planning problems. We hope our preliminary quantitative findings do provide some initial insights into practical algorithms. In our answer to your next question, we briefly discuss additional challenges that we might encounter when we shift towards constructing policy architectures and learning within more realistic frameworks, such as robotics.
>
> **Q2:** Analysis and construction for larger/more complex problems.
>
> **A2:** The theorems and constructions are general, so they apply to problems of any size. However, to build practical systems that plan in very large domains, additional methods for abstracting and factoring the planning problem, e.g., via hierarchical decomposition, will be crucial for efficient solutions.
>
> Neural network architecture: prior work [1] has demonstrated how different aggregation functions in graph neural networks carry different inductive biases. An intuitive example is that max-pooling over node features can act as an "existential" quantification, whereas mean-pooling can act as a majority voting. These considerations become crucial when tailoring solutions to specific problems.
>
> Of course, as the necessary circuits get more complex, the NN training may become more difficult. Also, more training data might be needed to provide something close to a "complete presentation" of the problem so that the RelNN can learn all regression rules in the domain.
>
> We'd like to emphasize that our work is an attempt to bring us a step closer to a comprehensive understanding of learning and planning with object-centric representations. We hope it will pave the way for future explorations.
>
> [1] Keyulu Xu et al. How Powerful are Graph Neural Networks? In ICLR, 2019.
>
> **Q3:** Applicability in real-world scenarios.
>
> **A3:** Thanks for bringing up this question. We will extend the discussion in future revisions. The proposed analysis and formulation does generalize directly to "real-world" settings (e.g., object features should be extracted from visual perception inputs) if we assume the input state can be represented in an object-centric, and discrete manner (a.k.a. as logic "predicates"). For example, [2] has implemented relational neural network architectures for BlocksWorld from visual inputs.
>
> It is challenging to generalize directly to scenarios where state variables and action parameters are truly continuous (e.g., in robotic manipulation tasks). The intuition is that, if actions have continuous parameters, there will be an infinite number of possible actions available at each state (depending on the choice of grasping poses, object placement poses, etc.). Therefore, the depth-first-search-based algorithm used in this paper will not work. Consequently, our policy compilation strategy will not work.
>
> Robotic manipulation problems can generally be more computationally complex than the discrete problems we studied because they can certainly contain those discrete problems as a sub-problem (e.g., a "robotic" blocks world). Meanwhile, any robotic planning problem that involves moving an articulated object with many links among many polygonal obstacles is PSPACE-complete [3]. Therefore, there are no efficient exact algorithms for these problems; many motion planning algorithms, for example, rely on sampling and discretization. However, how to better produce samples/discretize the space is generally hard. We believe that it is an interesting question to investigate similar "width"-like definitions in sampling and discretization-based continuous planning algorithms.
>
> [2] Richard Li, et al. Towards Practical Multi-Object Manipulation Using Relational Reinforcement Learning. In ICRA, 2020.
>
> [3] William Vega-Brown and Nicholas Roy. Task and Motion Planning is PSPACE-Complete. In AAAI, 2020.
>
> **Q4:** Paper organization, supplementary materials and links.
>
> **A4:** Thank you for suggestions. We will try our best to use the page in NeurIPS camera-ready if the paper gets accepted to include more intuitions, and definitely add links.
>
> **Q5:** The advantage of using a RelNN.
>
> **A5:** Thanks for raising this point. There are two major advantages of learning a RelNN. First, when domain is known (in STRIPS), the RelNN will have computation time O(depth of the circuit), due to its parallel execution nature. This is generally more efficient than running the classical planning algorithm. Second, there are scenarios when the domain is unknown to the agent, then RelNN is a method that supports learning policies from interactions or demonstrations. For example, consider an agent that is learning to play Minecraft, in which case we know that some crafting rules (e.g., in order to build an axe, we need wood planks and iron ore) have certain forms and thus can be serialized and have low width, but we still need to perform exploration in the environment to learn those rules.
>
> **Q6:** Serialization in mathematical optimization and dynamic programming.
>
> **A6:**  Serialization is strongly related to hierarchical decomposition, in which we assume we can solve some subpart of a problem without worrying about the rest of the problem. The connection to dynamic programming (in which we re-use solutions to subproblems in order to solve a bigger problem) is less clear to us, but it definitely bears additional thought!
>
> **Q7:** Applicability of the proposed formulation.
>
> **A7:** In this paper, we address the class of planning problems that can be formulated in atomic STRIPS. That is really only a small set of all interesting planning problems, and does not address continuous state/action spaces, stochasticity, partial observability, execution actions in parallel, etc.

---

> > ### Comment · Reviewer_7eTK · 2023-08-15
> > **Thanks**
> >
> > I appreciate the sincere and detailed explanations by the authors. Even though practicality concerns are addressed to some degree, it is not solid enough for me to raise my score at the moment.

---

### Official Review · Reviewer_aaQH · 2023-07-06

**Soundness:** 3 good
**Presentation:** 3 good
**Contribution:** 2 fair
**Rating:** 6
**Confidence:** 3

**Summary:**

This paper discusses the expressivity of classical relational architectures and its implication for solving planning problems. In particular, the authors identify characteristics of the environments that make planning hard. They prove a theorem that bounds the size of a network sufficient to solve the problem of planning perfectly, under some assumptions on the underlying domain. Finally, they show experimental results in two simple environments.

**Strengths:**

The paper is very clear and easy to follow. It's very convenient that you provide intuitive explanations of the derived theorems and explain them in an example environment. The motivation is clearly stated in the introduction. The problem is interesting and deserves a study. The presented results are novel and provide a deeper understanding of the capabilities of widely used architectures.

**Weaknesses:**

In my opinion, the greatest weakness of this paper is an insufficient discussion of the connection between the formulated theory and practical applications. I understand that the contribution of the work is mostly theoretical. Anyway, it would be nice to show how the presented theory should influence my design choices. For instance, I suggest providing a list of environments with a short explanation of their properties (finite/unbounded depth, serializability, width, theoretical bounds values, implications, etc.), even without experiments, even in the appendix.

**Questions:**

l.99 It's very convenient that you explain the discussed properties with an example. However, it would be even better if you briefly explained the meaning of actions. Although after reading I know what _clear_(A) means, explaining it at the beginning would be helpful. Also, generally, there is no precise description (even brief) of the BlocksWorld environment in the paper.

l.114 Please refer to a specific appendix.

l.124 I'm not sure if the provided intuition is correct. As far as I understand, it should be "A rule is optimally serializable if any optimal trajectory extended with an optimal continuation forms an optimal plan." That is, the intuition described in the paper suggests just an existence of $a_2$. Is that correct?

l.130 Does Theorem 3.1 have proof? Please, provide a reference.

l. 317 It would be nice to show how the presented theory should influence design choices when using relational networks. For instance, I suggest providing a list of environments with a short explanation of their properties (finite/unbounded depth, serializability, width, theoretical bounds values, implications, etc.), even without experiments, even in the appendix. The discussion provided in l.305-317 is a good starting point. The discussion in the appendix (again, no reference) is also nice, but I believe it can be extended. Notably, with more examples. This way the theory you provide could be much more impactful.

l.324 I like that you provide experiments both for theoretically bounded and unbounded depth. However, it's a shame that it turns out that unbounded depth is actually no bottleneck. In general, it's good to show such an environment, but here I'd like to see an experiment in an environment with truly unbounded depth and witness the growing success rate up to a practical bound. Because BlocksWorld actually requires little complexity, there is no experiment that links your theory with practice. How about BlocksWorld with a limited number of stacks? But it may fall in the category of "resource constraints", is that a problem? Actually, I do want here an environment with a _small_ but not _extremely small_ circuit.

l.334 What does it mean that a model has breadth 2? Does it mean that a layer has 2 neurons? I suppose not, please clarify that.

**Limitations:**

The limitations are briefly discussed in the conclusion. Negative societal impact is not an issue with this work.

---

> ### Author Rebuttal · Authors · 2023-08-08
>
> Thank you for your helpful comments!
>
> **Q1:** Connections to practical problems.
>
> **A1:** Thank you for the suggestion. We have included a new discussion of 6 popular problems that have been widely used in AI planning communities and discuss the circuit depth and breadth of them. We summarized their regression width, circuit depth, and circuit breadth in the following table. Please refer to our general response for more discussions (e.g., finding optimal plans vs. finding satisficing plans). These results can indeed be used in choosing neural network hyperparameters (e.g., the depth). We hope these discretized/simplified versions of practical problems can give us insight into their relative difficulty.
>
> | Problem     | Breadth   | Regression Width | Depth    |
> | --------    | --------  | ---------------- | -------- |
> | BlocksWorld | Constant  | 1 | Unbounded |
> | Logistics   | Constant  | 0 | Unbounded |
> | Gripper     | Constant  | 0 | Constant  |
> | Rover       | Constant  | 0 | Unbounded |
> | Elevator    | Constant  | 0 | Unbounded |
> | Sokoban     | Unbounded | N/A (not serializable) | Unbounded |
>
> **Q2:** Clarifications on the Blocks World environment and appendix references.
>
> **A2:** Thanks for the suggestion, we will include descriptions of the state representations in the BlocksWorld example, and put exact section numbers when we refer to appendices.
>
> **Q3:** L124, intuition about the goal regresion rule.
>
> **A3:** Thanks for catching that! Your intuition is correct and we will update the manuscript.
>
> **Q4:** Proof for Theorem 3.1.
>
> **A4:** Yes, the proof was attached in Appendix A.1. We will add an reference.
>
> **Q5:** Experiments showing the importance of truly unbounded depth.
>
> **A5:** Thanks for the suggestion! We totally agree that there should be an additional environment better showing the importance of unbounded depth.
>
> In addition to the "no resource constraint" feature you mentioned, there is another aspect of BlocksWorld that makes finding a satisficing plan easy: non-existence of deadends (as long as you keep moving objects down to the table, you will eventually make the target object clear). An environment is that does not have this structure is Logistics (or any graph path problem in a non-strongly-connected graph).
>
> In particular, we constructed a simple Logistics domain with only cities and trucks (no airplanes), and we construct the graph so that it's not strongly connected (by first sampling a directed tree and then add forward-connecting edges). Similar to other models trained in the environment, we train the policy network on problems with less than n=10 cities and test it on n=30 and n=50. The results are summarized below (with f(n)= n/5 + 1).
>
> | Model | n = 10 | n = 30 | n = 50 |
> | -------- | -------- | -------- | -------- |
> | RelNN(3, 2) | 1.0  | 0.30 | 0.23 |
> | RelNN(f(n), 2) | 1.0  | 1.0  |  1.0 |
>
> The results show that having an unbounded depth circuit is important in learning a generalizable policy in this domain. We will add code for reproducing this experiment to the released code.
>
> **Q6:** L334 Breadth.
>
> **A6:** Breadth means the maximum arity in the relational representation. Breadth 2 means that in the RelNN, there is a vector representation between each pair of objects. For the "number of neurons", we use 64 for all experiments, which should be sufficiently large for the particular domains we are considering.

---

> > ### Comment · Reviewer_aaQH · 2023-08-16
> >
> > Thank you for the response. Including the clarifications and the presented discussion of additional environments makes the paper stronger. I acknowledge the proposed changes by increasing my rating.

---

### Official Review · Reviewer_bz43 · 2023-07-09

**Soundness:** 4 excellent
**Presentation:** 2 fair
**Contribution:** 2 fair
**Rating:** 6
**Confidence:** 3

**Summary:**

The paper presents a mapping from tractable but incomplete algorithm classical planning into circuits that can be represented by a class of neural networks called relational neural networks RelNN including transformers architecture and graph neural networks. The paper is divided into two parts.

The first part presents an extension of Chen and Gomez [CG2007] that introduced a notion of width, in turn, related to the more practical notion of width proposed by Lipovetzky and Geffner [LG2012]. The change is about using regression search –not a big change given that as a theoretical algorithm– while keeping the same idea of keeping track of a reduced context.

The second part considers how to map policies based on those algorithms –given a set of regression rules– into a RelNN. The paper discusses the size of the required network in different cases. The best scenario is when the size of the NN doesn't grow linear with the size of the problems. Using the ideas of the first part, the paper shows how can that be expressed as a RelNN. Preliminary empirical results over a set of problems suggest a MLP is learning a policy that matches the ideas of the paper.

**Strengths:**

- Innovative combination of tractable planning with the expressivity of some NN. Serialization is a suggestive idea for a tractable fragment for planning.
- The idea of 'regression rule selector' might be hard to use in an effective symbolic algorithm. Framing it as learning is an innovative idea.


**Weaknesses:**

- The paper offers little insight into how depth and breadth behave in different problems. In particular, it's not clear how the complexity of the rules is related to that. Perhaps a better title for the current manuscript would be "Relational Neural Network can Solve some Planning Problems", as the paper does not help to identify what problems can be solved except by trial-and-error. That might be a good contribution, but then the tone of the paper might be different.
- The paper doesn't clarify whether the goal of the approach is to solve a single problem or problems from a domain.
	- For instance, the width of [LG2012] refers to problems, but then they discuss width across instances of a problem.
	- This might be easily fixed in the next discussion.
- Experiments cannot be replicated as we lack details.


**Questions:**

- What's the goal stack in the algorithms?
- How are the objects encoded in the RelNN?
	- What's the difference/similarity in the encoding of problems where the policy have bounded vs unbouded depth?
- Is the encoding the same for two problems where we interchange the name of two objects? For instance, rename A s B and B as A in a blocks world problem.
	- Might such renaming have any effect in the experiments?
- Are MLP per se part of the class of relational NN?
- How are the problems and outputs encoded in the MLP?
- The proof of 4.3 says that rules include \rho, any kind of FOL. Lemma 4.1 refers to  FOL but only restricts the number of variables.
	- How does D change with the complexities of the formulas?
- In thm 4.3, why does the breath only grows with max(B_r, max arity of predicates)?
	- Perhaps it's because the preconditions are evaluated once at a time, but the end of the proof says "\beta is required to store the input state". A state includes more than one predicate.

**Limitations:**

This is a very interesting line of work combining results in different areas. The theoretical arguments focus on the existence of a bound, there is little formal study or intuition of how the complexity grows across different problems. For instance, if conditions are free FOL, for a fixed number of variables and objects, then the resulting circuits can have many different sizes.

In general many intuitions are hidden in the proofs ideas and not in the main text. That's a possible reasonable decision, but sometimes the key intuitions are hidden there and not connected with the overall picture.

Lacking intuitions, the empirical study offer some light in this direction, but a reader in NeurIPS might need further details on how this was done.

Finally, I think this would be a better paper with a more clear distinction between problems and domains. Previous work using the notion of width tends to discuss it across a class of problems. In that setting, providing or obtaining a set of rules is more clear that for solving a single problem. The set of rules might be very different for subsets of problems. (For instance, blocks world instances where all the blocks start at the table). That framing would allow to establish connections with work in learning for generalized planning.

For that reason and others, this one is a very relevant reference:
Learning Sketches for Decomposing Planning Problems into Subproblems of Bounded Width. Dominik Drexler, Jendrik Seipp, Hector Geffner.
ICAPS 2022.
https://arxiv.org/abs/2203.14852

All in all, I learned a lot from the paper. That's a positive sign.

Minor point(s):
- why can serialization be incomplete? Perhaps explaining this might be useful: https://en.wikipedia.org/wiki/Sussman_anomaly

---

> ### Author Rebuttal · Authors · 2023-08-08
>
> Thank you for your helpful comments!
>
> **Q1:** Replication of experiments.
>
> **A1:** Thanks for the suggestion. We will extend Appendix C to include more details. Also, we have released an anonymized version of the code with the supplement (since NeurIPS rebuttal does not allow for URLs, please check our Appendix D for the link).
>
> **Q2:** What's the goal stack in the algorithms?
>
> **A2:** Recall that both BWD and GRS are depth-first search algorithms. Here, "goal stack" means all the `goal` (or `goal_set`) arguments in the function calling stack. We will clarify that in the revision.
>
> **Q3:** How are the objects encoded in the RelNN?
>
> **A3:** In RelNNs (e.g., graph neural networks), the state is represented as a graph (objects are nodes and relations are edges; possibly there will hyperedges for multi-ary relations). Therefore, the objects are encoded as node indices. For example, the input contains nullary features (a single vector for the entire environment), unary features (a vector for each object), binary features (a vector for each pair of objects), etc. They are usually represented as tensors.
>
> **Q4:** What's the difference/similarity in the encoding of problems where the policy has bounded vs unbounded depth?
>
> **A4:** We use the same encoding style for all problems (the graph-like relationship). Therefore, whether different problems have bounded or unbounded width primarily depends on the available goal regression rules, how these rules can be serialized, and the width of the rules.
>
> **Q5:** Will changing of object names affect results?
>
> **A5:** No. In all our experiments, objects do not have names / identity information exposed to the neural network. This is related to the "permutation-invariance" of RelNNs, which means that any permutation of the input object set will not change the model performance.
>
> **Q6:** Is MLP a RelNN?
>
> **A6:** Yes, MLP is a "degenerate" RelNN with breadth 0. That is, the entire state is represented as a single vector.
>
> **Q7:** How are the problems and outputs encoded in the MLP?
>
> **A7:** We assume you meant to ask how problems and outputs are encoded in the relational neural networks. Here, the inputs to the neural network are tensor representations of the current state (e.g., $N\times N$ matrices for a predicate of arity 2, where $N$ is the number of objects). Since we assume that the goal contains only one single atom, goals can be encoded as tensors too. For example, suppose that our goal is on(obj1, obj2), then it can be encoded as a matrix where only the entry corresponding to the pair (obj1, obj2) is 1; all other entries are 0.
>
> **Q8:** Depth bounds in Thm 4.3.
>
> **Q8:** Thank you for pointing this out! We plan to add more intuition for both Lemma 4.1 and later theorems. In short, there is a constructive proof for using RelNNs to realize FOL formulas. The breadth B is the number of variables in FOL, the depth D is the number of "nested quantifiers" in an FOL formula. For example, the formula `exists x. forall y. p(x, y)` needs two layers while `exists x. forall y. forall z. p(x, y, z)` needs 3. We will also clarify that the FOL formulas in $\rho$ should be realizable by a RelNN with constant depth and breadth.
>
> **Q9:** Breadth bounds in Thm 4.3.
>
> **A9:** The intuition is that, if there are multiple preconditions for different rules, they can be evaluated "in parallel" and do not lead to larger breadth than individual rules: recall that breadth is actually the "arity" of edges in the graph. If we have many rules, we can increase the "hidden dimension" for each edge representation. Similarly, for state encoding, if we have multiple (say, 8) predicates (say, of arity 3), we can represent them by a tensor of size $N \times N \times N \times 8$, where $N$ is the number of objects in the state. Here, we say the breadth of this representation is 3, because it needs $O(N^3)$ memory to encode.
>
> **Q10:** Limitations on missing intuitions behind proofs and how complexity grows across different environments.
>
> **A10:** Thank you for suggesting these. We will try our best to use the additional page in NeurIPS camera-ready version, if the paper gets accepted, to include more intuitions behind theorems and definitions. Based on your suggestion, we also added ...
>
> **Q11:** Distinction between problems and domains.
>
> **A11:** Thank you for the great suggestion! We will definitely incorprate this suggestion in our future revision. Concretely, we plan to add a definition for "problem classes" composed of problems that share the same domain (e.g., transition model) but vary in $s_0$ and goals. And then, we will discuss width of problem classes and policy complexities for problem classes.
>
> **Q12:** References.
>
> **A12:** Thank you for the references! We will cite the Drexler et al. paper on learning sketches: their idea of sketch is definitely related to our goal regression rule. The key differences are: their skeches can be viewed as high-level, feed-forward policies. By contrast, in our conditional rules, there might be multiple rules applicable at the current state, so search is still required. We will also use the Sussman anomaly example to illustrate the idea of incompleteness.'

---

> > ### Comment · Reviewer_bz43 · 2023-08-14
> > **thank you**
> >
> > Thank you for the thoughtful response. I'm satisfied. I think the analysis per domain can be quite insightful.
> > In your response, I see the mention of parameters on the structure of the formulas. In my opinion, this suggests that the encoding, decoding, and parameters should be made clear as the paper progress.
> > "Enconding graph" is sometimes abused in the literature, leaving the reader with no idea of about affects the size of the network and the expressivity.

---

> > > ### Author Response · Authors · 2023-08-14
> > > **Thank you for your feedback**
> > >
> > > Thank you for your response and constructive suggestions! We will incorporate a more clear definition of the representation of the input and the output, especially for multi-ary relations. We will also add details for the computation in RelNNs. Thank you again for your valuable feedback.

---

### Author Rebuttal · Authors · 2023-08-08

We thank all reviewers for their thoughtful and constructive comments. Some of the reviewers have suggested applying the theorems and analyses developed in this paper to a broader set of problems. Here, we list 6 popular problems that have been widely used in AI planning communities and discuss their circuit depth and breadth. We summarize our results in the following table.


| Problem     | Breadth   | Regression Width | Depth    |
| --------    | --------  | ---------------- | -------- |
| BlocksWorld | Constant  | 1 | Unbounded |
| Logistics   | Constant  | 0 | Unbounded |
| Gripper     | Constant  | 0 | Constant  |
| Rover       | Constant  | 0 | Unbounded |
| Elevator    | Constant  | 0 | Unbounded |
| Sokoban     | Unbounded | N/A (not serializable) | Unbounded |

Note that here we are limiting our discussion to the case where the goal of the planning problem is a single atom. In summary, most of these problems have a constant breadth (i.e., the regression width of the corresponding problem is constant) except for Sokoban. Most problems have an unbounded depth: that is, the depth of the circuit will depend on the number of objects (e.g., the size of the graph or the number of blocks. The constant breadth results can be proved by construction: list all regression rules in the domain. We have analyzed the complexity of BlocksWorld and Logistics in Appendix A.6; all other problems can be proved in a similar way. The problem gripper has a constant depth under single-atom goal because it assumes the agent can move directly between any two rooms; therefore, no pathfinding is required. Sokoban has an unbounded breadth even for single-atom goals because if there are multiple boxes blocking the way to a designated position, the order to move the boxes can not be determined by simple rules.

For problems in this list, when there are multiple goals, usually the goals are not serializable (in the optimal planning case). This can possibly be addressed by introducing "super predicates" that combine two literals. For example, if two goal atoms p(x) and q(x) are not serializable, we can introduce a new predicate p_and_q(x), and rewrite all operators with this new super predicate. Appendix A.5 provides more discussion and examples. This will make the problem serializable but at the cost of significantly enlarging the set of predicates (exponentially with respect to the number of goal atoms).

If we only care about satisficing plans, for Logistics, Gripper, Rover, and Elevator, there exists a simple serialization for any conjunction of goal atoms (basically achieving one goal at a time).

These theoretical analyses can be used in determining the size of the policy circuit needed for each problem. For example, one should use a RelNN of breadth $(k + 1) \cdot \beta$, where $k$ is the regression width and $\beta$ is the max arity of predicates. For problems that have unbounded depth, usually, the depth of the circuit grows in $O(N)$, where $N$ is the number of objects in the environment (e.g., in Elevator, the number of floors).

---

> ### Comment · Reviewer_TK6j · 2023-08-16
>
> Thank you for sharing this. Please add this to the paper or appendix. This is really helpful.

---

### Author Response · Authors · 2023-08-21
**Thank You**

Dear reviewers and area chairs,

As the reviewer-author discussion phase is coming to an end, we would like to say thank you, and that we truly appreciate your constructive feedback. We are glad that our rebuttal and additional results have addressed the concerns from all reviewers, and we will definitely incorporate the suggested changes and new discussions in our next version.

Authors

---

### Decision · Program_Chairs · 2023-09-21

**Decision:**

Accept (spotlight)

**Comment:**

The reviewers found the paper to make an interesting and novel contribution. Please take the critical feedback into account when preparing the camera ready version.